# Rank-guided Diffusion for Noise Few-Shot Learning

Zelei Wu [* 1]   Kun Zhou [* 2]   Xulun Ye [1]   Yifan Mei [1]   Jie Hong [1]   Jieyu Zhao [1]

## Abstract

In real-world Few-Shot Learning (FSL), support sets are quickly constructed and inevitably contain noisy samples. With limited examples per class, even a single noisy instance can distort class distributions, cause prototype drift, and reduce generalization. Existing methods mostly assume clean data or require large-scale statistics, which are impractical in FSL's data-scarce setting. We find that clean samples in semantic feature space lie in low-rank subspaces, while noisy samples cause rank anomalies disrupting this structure. To address this, we propose a differentiable low-rank approximation that estimates the intrinsic rank of the support set and detects anomalous noisy samples. Building on this, a rank-guided diffusion process generates high-quality replacements under low-rank constraints, reconstructing a clean, consistent support set for improved robustness. This low-rank guided approach effectively mitigates prototype drift and significantly reduces errors under noise levels up to 40% across MiniImageNet, TieredImageNet, and other noisy benchmarks, demonstrating the power of low-rank geometry for noise detection and correction in FSL. Our source code is available at https://github.com/wuzelei123/CRDProto

## 1. Introduction

Few-Shot Learning (FSL) aims to recognize novel categories from only a handful of labeled examples and has achieved remarkable progress through diverse strategies, including semantic enhancement (Zhang et al., 2024b), adversarial defense (Liu et al., 2024), loss landscape smoothing (Zou et al., 2024), diffusion-based meta-learning (Zhang et al., 2024a), feature regularization (Zhu et al., 2024), do-main adaptation (Wang et al., 2024), as well as the adoption of large-scale foundation models such as CLIP and GPT-3 (Brown et al., 2020a). Despite these advances, the reliability of FSL systems in realistic settings remains fundamentally constrained by the presence of noisy samples in the support set.

In practical deployments, support sets are rarely curated under controlled conditions. Labels may be obtained from web-crawled data, weak supervision, or non-expert annotators, where noise is inevitable. Crucially, unlike standard supervised learning, the few-shot regime lacks sufficient sample redundancy to absorb or average out corrupted labels. Even a single noisy instance can significantly distort class prototypes, shift decision boundaries, and induce systematic bias. This sensitivity makes robustness to noisy samples a central yet unresolved challenge in few-shot learning.

Several recent methods attempt to mitigate noisy-label issues in FSL, such as selecting high-discrepancy samples (Shao et al., 2024a) or jointly denoising images and labels (Zhang et al., 2023a). However, these approaches typically assume partial label correctness or rely on inter-sample discrepancies to identify noise. Such assumptions often break down when the support set is both extremely limited and severely corrupted. More importantly, most existing methods treat noisy samples as independent outliers and largely overlook the geometric disruption that noise introduces into intra-class feature distributions.

In this work, we model noise from a geometric and structural perspective. We make the key observation that, in semantic feature spaces—particularly those induced by modern pretrained vision-language models—clean samples from the same class tend to reside in an approximately low-rank linear subspace. In contrast, noisy samples manifest as rank anomalies that violate this structural consistency. This phenomenon is empirically consistent across datasets and feature extractors, and becomes especially pronounced in semantically aligned representations. As shown in Fig. 1 (a) and (b), CLIP features exhibit a sharply decaying singular value spectrum, with energy concentrated in a few dominant components: the first principal component alone explains nearly 40% of the total variance, compared to approximately 12% for ResNet12. This compact structure, inherited from CLIP's image–text contrastive pretraining, provides a prin-

---

[*]Equal contribution  [1]School of Artificial Intelligence, Ningbo University, Ningbo, China [2]School of Architecture & Urban Planning, Shenzhen University, Shenzhen, China. Correspondence to: Xulun Ye <yexlwh@163.com>.

*Proceedings of the 43rd International Conference on Machine Learning*, Seoul, South Korea. PMLR 306, 2026. Copyright 2026 by the author(s).

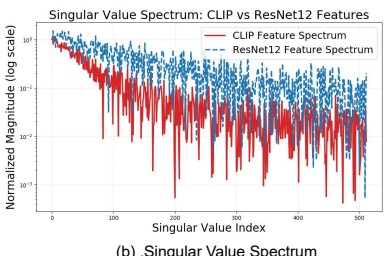
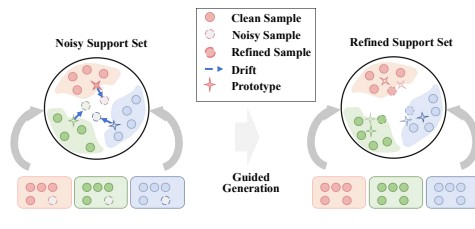

(a) Component Energy        (b) .Singular Value Spectrum        (c) Motvation

*Figure 1.* (a) Motivation: The solid colored circles represent clean samples in the support set, while dashed gray circles indicate noisy samples that may cause the aggregated prototype to drift away from the true category center. By employing guided generation to reposition deviated noisy samples back into their correct category space, the refined support set ensures the prototype remains accurately aligned within each class. (b) Principal Component Energy Distribution and (c) Singular Value Spectrum: Comparative analysis between CLIP and ResNet12 features shows that CLIP exhibits stronger low-rank properties, with its first principal component explaining nearly 40% of the total variance (vs. 12% for ResNet12) and a sharply decaying singular value spectrum, indicating a more compact representation in lower-dimensional subspaces.

cipled foundation for low-rank geometry–based noise modeling.

Building on this observation, we develop a low-rank representation framework that leverages rank deviation as a geometric prior for identifying noisy samples in few-shot support sets. However, directly removing or projecting out noisy samples based on rank information can lead to irreversible information loss and prototype collapse—an undesirable effect in few-shot scenarios where each sample carries substantial semantic value. To address this issue, we introduce a rank-guided diffusion strategy that performs gradual structural correction within each class's low-rank subspace. Rather than treating diffusion as a purely generative process, we view it as a structured uncertainty propagation mechanism that iteratively refines corrupted support features while preserving intra-class semantic diversity. Specifically, detected noisy samples are replaced by synthesized features constrained to the intrinsic low-rank subspace of their corresponding class, resulting in a refined support set that is both geometrically consistent and semantically faithful (Fig. 1 (c)). This design effectively mitigates prototype drift caused by noisy samples and enhances the robustness and generalization of few-shot learning systems.

The main contributions of this paper are summarized as follows:

- We identify low-rank structural consistency as a robust geometric prior for modeling noisy samples in few-shot learning, particularly in pretrained semantic feature spaces.

- We propose a rank-guided diffusion framework that performs gradual structural correction of corrupted support samples under principled low-rank constraints.

- We provide an analytical perspective on the proposed rank-guided diffusion process, together with a re-

stricted contraction analysis of the orthogonal residual under low-rank guidance. Extensive experiments further demonstrate that the proposed framework improves robustness under noisy-label few-shot learning settings.

## 2. Related Work

### 2.1. Few-shot Learning

Few-shot learning (FSL) aims to recognize novel categories from extremely limited labeled data and has been studied under several representative paradigms, including metric-based learning, data augmentation, and meta-learning. Metric-based methods classify queries by measuring similarity to support samples in a learned embedding space. Representative approaches include Prototypical Networks (Snell et al., 2017), which represent each class by the mean feature of its support samples, Matching Networks (Vinyals et al., 2016a), which perform instance-level similarity matching, and Relation Networks (Sung et al., 2018), which introduce a learnable relation module to model pairwise relationships. These methods achieve strong performance under clean few-shot settings. Data augmentation methods alleviate data scarcity by increasing sample diversity at the image level (Qin et al., 2020; Xu et al., 2021) or feature level (Schwartz et al., 2018; Xian et al., 2019; Li et al., 2020b). Meta-learning approaches aim to enable fast adaptation by learning transferable meta-knowledge, including gradient-based methods such as MAML (Finn et al., 2017; Antoniou et al., 2018) and optimization- or transfer-based strategies such as MetaOptNet (Lee et al., 2019b) and Meta-Transfer Learning (Lee et al., 2019a).

### 2.2. Few-Shot Learning with Foundation Models

Recent advances in foundation models, particularly vision–language models such as CLIP (Radford et al., 2021),

have significantly advanced few-shot learning by enabling effective knowledge transfer under limited supervision. Building upon CLIP, DeIL (Shao et al., 2024b) improves robustness under limited and noisy supervision through a two-stage inference mechanism. CO3 (Shao et al., 2024a) integrates multiple foundation models, including CLIP, DINO (Zhang et al., 2022), GPT-3 (Brown et al., 2020b), and DALL-E (Ramesh et al., 2021), to jointly perform label correction, data augmentation, and multi-modal fusion without fine-tuning. Similarly, CaFo (Zhang et al., 2023b) proposes a "Prompt–Generate–Cache" pipeline to fuse predictions from CLIP and DINO, while Proto-CLIP (Chen et al., 2023) aligns image and text prototypes to enhance classification under low-data settings.

# 3. Methods

## 3.1. Motivation & Idea

In few-shot learning tasks, we observe that samples from the same class typically lie within an approximately low-rank linear subspace in a semantically rich feature space such as CLIP, exhibiting strong geometric consistency. This low-rank structure not only reflects the compactness within the class but also provides a stable geometric prior for modeling the support set, serving as an important criterion for assessing sample purity. However, real-world support sets often contain noisy samples such as label errors, occlusions, complex backgrounds, and incorrectly labeled irrelevant images. These samples deviate from both the class-specific low-rank subspace and the semantic center, disrupting the structural consistency of the support set and thereby affecting classification reliability.

Formally, let $X^{(c)} \in \mathbb{R}^{d \times K}$ denote the feature matrix of class $c$ in the support set. In clean few-shot episodes, samples from the same class typically concentrate around a low-dimensional semantic structure. Rather than assuming an exact rank-one model, we empirically observe that the singular values of $X^{(c)}$ decay rapidly, so that most of its energy is captured by a small number of dominant components. Equivalently, the class-wise feature matrix exhibits a low effective rank.

When concatenating features from all $N$ classes, the entire support matrix is given by

$$X = [X^{(1)}, X^{(2)}, \ldots, X^{(N)}] \in \mathbb{R}^{d \times (NK)}. \qquad (1)$$

When the class-specific subspaces are reasonably separated, the clean support set admits a blockwise low-dimensional structure: each $X^{(c)}$ is well approximated by a low-rank matrix, and the dominant subspaces of different classes exhibit limited overlap. In this sense, the support set is characterized not by an exact algebraic rank constraint, but by a structured low-dimensional geometry. In practice, this geometry is easily disrupted by label noise, background clutter, occlusion, or atypical viewpoints. Such noisy samples introduce additional directions that are weakly aligned with the dominant class subspace, thereby increasing the effective rank, flattening the singular-value spectrum, and weakening inter-class separation. This structural degradation undermines prototype estimation and reduces the reliability of metric-based classification. Our method is built on this empirical observation: clean support features tend to exhibit concentrated spectral energy, whereas noisy samples induce deviations in directions outside the dominant class-specific subspace.

## 3.2. Noise Sample Detection via Low-Rank Structural Consistency

As discussed in the previous section, real-world few-shot support sets often deviate from an ideal low-rank structure due to label errors, occlusions, complex backgrounds, or unusual viewpoints. Such noisy samples increase the rank of class-specific feature matrices and compromise the approximate orthogonality between different class subspaces, which can undermine prototype-based computations and low-dimensional representations. To mitigate this, we propose a noise detection method based on low-rank structural modeling, using intra-class geometric consistency as the main structural prior.

Let $X^{(c)} \in \mathbb{R}^{d \times n}$ denote the feature matrix for class $c$. We decompose $X^{(c)}$ into a low-rank component $L$, capturing the dominant intra-class structure, and a sparse component $S$, representing deviations caused by noise:

$$X^{(c)} = L + S. \qquad (2)$$

A sample is considered anomalous if its corresponding column in $S$ has significant magnitude.

To measure geometric deviation, we perform a singular value decomposition $L = U\Sigma V^\top$ and construct a class-specific subspace using the top-$k$ singular vectors $U_k$. The projection residual for each sample is then

$$e_i = \|f_i - U_k U_k^\top f_i\|_2^2, \qquad (3)$$

which quantifies how much the sample deviates from the dominant intra-class geometry. While this residual effectively captures structural inconsistencies, it may fail for samples that are visually plausible but semantically inconsistent. To address this, we introduce a lightweight semantic consistency check within the support set, comparing each feature $f_i$ to a class prototype $y_c$ in a shared embedding space:

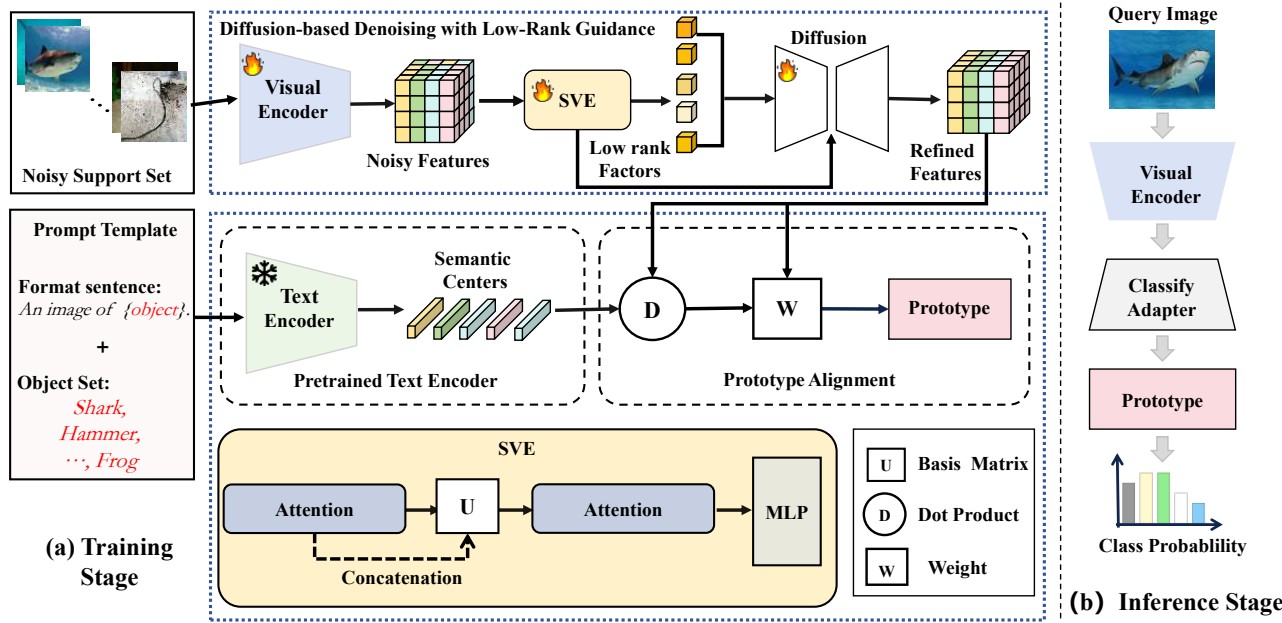

*Figure 2.* Framework of our proposed rank-guided diffusion for noisy few-shot learning. (a) Training stage: Given a noisy support set and its corresponding object set, we employ pre-trained visual and text encoders to extract noisy visual features and text embeddings. These noisy features are then processed by our Singular Value Embedding (SVE) module to derive low-rank factors. The SVE module is further integrated into the diffusion model to guide the generation process, producing refined visual features. Simultaneously, the refined features interact with text embeddings to enhance their discriminative power, which are then weighted into the prototype via our prototype alignment module to obtain the most representative visual features. (b) Inference stage: We use the learned visual encoder to extract visual feature from a query image and we utilize a classify adapter to make a prediction with the learned prototype, yielding the final class prediction.

$$s_i = \frac{f_i^\top y_c}{\|f_i\| \cdot \|y_c\|}. \tag{4}$$

Lower similarity indicates potential semantic inconsistency or label noise. Importantly, the semantic signal is used only as an auxiliary cue for support set correction; the framework primarily relies on relative semantic agreement within the class and is agnostic to the specific choice of semantic encoder. A sample is flagged as noisy only when both geometric and semantic criteria are violated, ensuring that no single cue dominates the detection.

Traditional low-rank decomposition methods, such as hard singular value truncation or rank-constrained projections, are inherently non-differentiable. Discrete subspace selection and hard thresholding introduce discontinuities in the mapping from $X^{(c)}$ to $L$, preventing gradients from propagating and making end-to-end training infeasible. To overcome this, we introduce a differentiable Singular Value Embedding (SVE) module. SVE approximates the top-$k$ subspace via power iteration and applies a smooth soft-thresholding to the singular values, producing a continuous, differentiable mapping from the input features to their low-rank approximation. Orthogonality regularization

$$\mathcal{L}_{\mathrm{orth}} = \|U_k^\top U_k - I\|_F \tag{5}$$

is applied to maintain subspace stability and integrity. This design allows $L^{(c)}$ to be differentiable with respect to $X^{(c)}$, enabling joint optimization of geometric reconstruction, sparsity, and structural regularization in an end-to-end fashion:

$$\min_{\theta,\phi} \|X^{(c)} - L^{(c)}\|_F^2 + \lambda\|S^{(c)}\|_1 + \gamma_{orth}\mathcal{L}_{\mathrm{orth}}. \tag{6}$$

By integrating geometric and semantic cues (with semantics applied only for support set refinement) into a differentiable low-rank decomposition, the proposed method effectively identifies noisy samples while preserving intra class coherence and inter-class separability, providing a robust foundation for downstream few-shot classification and prototype-based learning.

### 3.3. Rank-Guided Diffusion for Few-Shot Denoising

To incorporate the class-specific low-rank structure into feature refinement, we combine the subspace extracted by

the differentiable Singular Value Embedding (SVE) module with diffusion-based generation.

Let the estimated low-rank subspace of class $c$ be

$$\mathcal{U}(c) = \mathrm{span}(U(c)), \qquad U(c) \in \mathbb{R}^{d \times r}, \qquad (7)$$

where $U(c)$ is obtained from the support set using the SVE module. Concretely, SVE approximates the dominant subspace via power iteration and singular-value shrinkage, yielding a smooth and differentiable mapping from the feature matrix to its low-rank approximation $L^{(c)}$. To stabilize the extracted basis, we further impose the orthogonality regularization

$$\mathcal{L}_{\mathrm{orth}} = \|U(c)^\top U(c) - I\|_F. \qquad (8)$$

This design enables end-to-end optimization of geometric reconstruction, sparsity, and structural regularization.

To inject the low-rank information into diffusion, we define the projection residual of a sample $z$ with respect to the subspace $\mathcal{U}(c)$ as

$$R(z) = \|(I - P_{U(c)})z\|_2^2, \qquad P_{U(c)} = U(c)U(c)^\top, \quad (9)$$

and modify the reverse diffusion update by adding a low-rank guidance term:

$$z_{t-1} = \frac{1}{\sqrt{\alpha_t}} \left( z_t - \frac{1 - \alpha_t}{\sqrt{1 - \bar{\alpha}_t}} \epsilon_\theta(z_t, t) \right) - \eta_t \nabla_{z_t} R(z_t) + \sigma_t \xi. \qquad (10)$$

Since

$$\nabla_z R(z) = 2(I - P_{U(c)})z, \qquad (11)$$

the low-rank guidance term acts only on the orthogonal complement of the estimated class-specific subspace $\mathcal{U}(c)$. To make this explicit, we decompose

$$z_t = P_{U(c)}z_t + (I - P_{U(c)})z_t, \qquad (12)$$

and define the orthogonal residual as

$$e_t := (I - P_{U(c)})z_t. \qquad (13)$$

Then the regularization energy can be written as

$$R(z_t) = \|e_t\|_2^2, \qquad (14)$$

showing that the proposed guidance explicitly penalizes only the component of $z_t$ that deviates from the dominant class-specific subspace. Therefore, the reverse update should be interpreted not as a global contraction over the entire feature space, but as a targeted correction mechanism on the off-subspace component. This selective suppression is particularly desirable in noisy few-shot settings, as it encourages samples to move closer to the dominant class geometry while comparatively preserving informative variations within the estimated subspace.

**Energy-based interpretation.** The low-rank guided reverse process can be interpreted as denoising under an additional structural energy

$$E_{\mathrm{LR}}(z) = \lambda \|(I - P_{U(c)})z\|_2^2, \qquad (15)$$

which penalizes deviations from the estimated class-specific subspace. Under this view, the standard diffusion term accounts for likelihood-based denoising, while the additional low-rank guidance biases the reverse trajectory toward structurally consistent representations. We emphasize that this is an energy-based interpretation rather than a claim of exact Bayesian posterior sampling.

**Restricted contraction analysis.** Motivated by the above decomposition, we analyze the guided reverse process through the orthogonal residual $e_t = (I - P_{U(c)})z_t$ rather than the full feature $z_t$.

**Proposition 3.1** (Contraction of the orthogonal residual).
*Assume that the effective error induced by score prediction and stochastic sampling in the orthogonal complement is bounded in second moment, i.e.,*

$$\mathbb{E}\big[\|(I - P_{U(c)})\Delta_t\|_2^2\big] \leq C_t, \qquad (16)$$

*where $\Delta_t$ denotes the effective reverse-step error at time step $t$. Then, for a sufficiently small guidance strength $\eta_t$, there exists $\rho_t \in (0, 1)$ such that*

$$\mathbb{E}\|e_{t-1}\|_2^2 \leq \rho_t \, \mathbb{E}\|e_t\|_2^2 + C_t', \qquad (17)$$

*where $C_t'$ collects the residual error caused by imperfect denoising and injected Gaussian noise. Consequently,*

$$\mathbb{E}\|e_t\|_2^2 \leq \left( \prod_{s=t+1}^{T} \rho_s \right) \|e_T\|_2^2 + \sum_{j=t+1}^{T} \left( \prod_{s=t+1}^{j-1} \rho_s \right) C_j'. \qquad (18)$$

*In particular, when the residual terms remain small, the orthogonal residual decreases geometrically toward a bounded neighborhood of the estimated class-specific subspace.*

This result does not claim global strong-convex convergence over the full feature space. Instead, it establishes a more targeted property that is sufficient for our method: the low-rank guidance progressively suppresses off-subspace deviations while preserving the dominant in-subspace structure.

In practice, noisy support features $\mathbf{f}_i \in \mathcal{S}_{\mathrm{noise}}$ are replaced by diffusion-generated counterparts $\tilde{\mathbf{f}}_i$ that are better aligned with $\mathcal{U}(c)$, yielding a refined support set

$$\tilde{\mathcal{S}} = \mathcal{S}_{\mathrm{clean}} \cup \{\tilde{\mathbf{f}}_i\}. \qquad (19)$$

The refined support set exhibits improved intra-class coherence and more stable low-dimensional geometry, providing a stronger foundation for subsequent prototype construction and few-shot classification.

## 3.4. Semantic-Center Weighted Prototype Estimation

To incorporate semantic priors into prototype estimation, we assume that the class prototype $p$ should remain close not only to the support features but also to the semantic center $C_s$. This leads to the following MAP estimation objective:

$$p_{\text{MAP}} = \arg\min_p \left( \frac{\beta}{2} \sum_{i=1}^N \|f_i - p\|_2^2 + \frac{\gamma_{map}}{2} \|p - C_s\|_2^2 \right), \tag{20}$$

where $\beta$ controls the contribution of the support features and $\gamma$ determines the strength of the semantic prior. Solving this objective yields

$$p_{\text{MAP}} = \frac{\beta \sum_{i=1}^N f_i + \gamma C_s}{\beta N + \gamma}. \tag{21}$$

This estimator can be interpreted as a convex combination of the empirical feature mean and the semantic center, balancing visual evidence and semantic prior knowledge.

To enhance robustness against residual outliers, we further introduce sample-level importance weights and generalize the MAP estimator to a weighted form. Specifically, we define the final prototype as

$$p_{\text{final}} = \frac{\beta \sum_{i=1}^N w_i f_i + \gamma C_s}{\beta \sum_{i=1}^N w_i + \gamma}, \qquad w_i \geq 0. \tag{22}$$

In our implementation, the weights are normalized to satisfy $\sum_{i=1}^N w_i = 1$, so Eq. (22) becomes

$$p_{\text{final}} = \frac{\beta \sum_{i=1}^N w_i f_i + \gamma C_s}{\beta + \gamma}. \tag{23}$$

This formulation preserves the role of the semantic center while reducing the influence of unreliable samples through adaptive weighting.

The weight $w_i$ is determined by the agreement of sample $f_i$ with both the geometric structure and the semantic prior. We first define the geometric and semantic consistency scores as

$$s_i^{\text{geo}} = \exp\left( -\beta_{geo} \|f_i - p_{\text{init}}\|_2^2 \right), \tag{24}$$

$$s_i^{\text{sem}} = \exp\left( -\gamma_{sem} \|f_i - C_s\|_2^2 \right), \tag{25}$$

where $p_{\text{init}}$ denotes the initial prototype estimated from the refined support set. The final sample weight is then computed as

$$w_i = \frac{\alpha_{mix} s_i^{\text{sem}} + (1 - \alpha_{mix}) s_i^{\text{geo}}}{\sum_{j=1}^N \left[ \alpha_{mix} s_j^{\text{sem}} + (1 - \alpha_{mix}) s_j^{\text{geo}} \right]}. \tag{26}$$

This weighting mechanism downweights samples that are inconsistent with either the low-rank geometry or the semantic center, thereby stabilizing prototype estimation in noisy support sets.

The entire process is differentiable and can be optimized end-to-end. When all samples receive equal weight, i.e., $w_i = 1/N$, Eq. (22) reduces to the unweighted MAP estimator in Eq. (21).

## 4. Experiment

### 4.1. Datasets

This study conducts experiments on two classic few-shot learning benchmark datasets: MiniImageNet(Vinyals et al., 2016b) and TieredImageNet(Sun et al., 2019), both containing 84×84 pixel images. Specifically, MiniImageNet consists of 64 training classes, 16 validation classes, and 20 testing classes, totaling 60,000 images. In comparison, TieredImageNet adopts a more fine-grained categorization with 351 training classes, 97 validation classes, and 160 testing classes, comprising approximately 780,000 images in total.Outlier noise is simulated by injecting samples from unrelated external classes into the support set, introducing out-of-distribution instances that reflect real-world contamination from unseen categories and Label-swap noise is generated by randomly reassigning a proportion of samples within the same dataset to different classes, simulating mislabeled instances.

### 4.2. Implementation Details

#### 4.2.1. NETWORK DETAILS

We propose **CRDProto**, a few-shot classification model that integrates prototype learning with a frozen pre-trained feature extractor. The model utilizes this extractor to extract high-dimensional visual features of dimension $D$, keeping the parameters frozen to ensure semantic consistency. To model inter-class relationships within the support set, the extracted features are passed through a lightweight Transformer module composed of $L$ layers and $N$-head self-attention. Learnable class tokens and positional embeddings are incorporated to enhance the expressiveness of prototype representations. The model introduces a pair of learnable orthogonal projection matrices to compress and reconstruct features before and after the Transformer: $W_{\text{in}} \in \mathbb{R}^{D \times d}$ projects the original features into a lower-dimensional subspace, and $W_{\text{out}} \in \mathbb{R}^{d \times D}$ reconstructs them back to the original space, effectively modeling low-rank feature representations. Furthermore, a residual enhancement mechanism stabilizes low-rank representations by concatenating the original support features with their prototypes and adding them to the Transformer output, mitigating gradient vanishing and preserving discriminative power.During prototype generation, the model derives class prototypes either directly from the class tokens or by aggregating transformed support features (e.g., mean pooling), depending on the architecture. To improve robustness in open-set scenarios, **CRDProto**

| Model | 0% | | 20% | | 40% | | 60% | |
|---|---|---|---|---|---|---|---|---|
| | Mini | Tiered | Mini | Tiered | Mini | Tiered | Mini | Tiered |
| MAML(ICML'17) | 63.21 ± 0.18 | 63.90 ± 0.19 | 57.35 ± 0.19 | 58.14 ± 0.19 | 50.00 ± 0.19 | 51.11 ± 0.20 | 40.90 ± 0.17 | 42.01 ± 0.20 |
| Vanilla ProtoNet(NIPS'17) | 68.18 ± 0.16 | 71.42 ± 0.18 | 63.92 ± 0.17 | 67.58 ± 0.19 | 57.07 ± 0.18 | 60.97 ± 0.20 | 46.99 ± 0.20 | 50.29 ± 0.21 |
| Baseline++(ICLR'19 ) | 67.85 ± 0.16 | 71.22 ± 0.18 | 63.49 ± 0.17 | 67.07 ± 0.19 | 56.84 ± 0.18 | 60.64 ± 0.20 | 46.96 ± 0.19 | 50.07 ± 0.21 |
| RNNP(WACV'21) | 68.17 ± 0.16 | 71.23 ± 0.18 | 63.80 ± 0.17 | 67.29 ± 0.19 | 56.97 ± 0.18 | 60.83 ± 0.20 | 46.92 ± 0.20 | 50.09 ± 0.21 |
| TraNFS(CVPR'22) | 68.11 ± 0.17 | 71.13 ± 0.19 | 64.96 ± 0.18 | 67.93 ± 0.20 | 59.03 ± 0.20 | 62.39 ± 0.22 | 47.69 ± 0.22 | 51.82 ± 0.23 |
| SemFew-Trans(CVPR'24) | 86.49 ± 0.50 | 89.89 ± 0.52 | 81.32 ± 0.46 | 84.43 ± 0.51 | 70.22 ± 0.56 | 73.39 ± 0.61 | 56.69 ± 0.54 | 58.82 ± 0.51 |
| MetaDiff(AAAI'24) | 81.21 ± 0.56 | 86.31 ± 0.62 | 75.32 ± 0.62 | 78.34 ± 0.62 | 69.27 ± 0.77 | 73.32 ± 0.19 | 53.22 ± 1.23 | 54.56 ± 0.92 |
| SemFew(CVPR'24) | 86.41 ± 0.56 | 89.89 ± 0.52 | 80.32 ± 0.66 | 83.25 ± 0.60 | 74.27 ± 0.73 | 76.32 ± 0.69 | 60.27 ± 0.56 | 62.34 ± 0.66 |
| FedFSL-CFRD(AAAI'25) | 72.15 ± 0.98 | 78.95 ± 1.28 | 62.21 ± 1.22 | 65.03 ± 1.56 | 58.27 ± 1.12 | 53.32 ± 1.19 | 46.69 ± 1.22 | 50.74 ± 1.22 |
| DETA(ICCV'2023) | 84.34 ± 0.33 | 80.39 ± 0.32 | 73.16 ± 0.31 | 71.68 ± 0.34 | 67.75 ± 0.32 | 64.79 ± 0.33 | 51.88 ± 0.32 | 50.38 ± 0.31 |
| DETA++(TPAMI) | 89.34 ± 0.36 | 85.39 ± 0.42 | 78.16 ± 0.42 | 76.68 ± 0.11 | 72.75 ± 0.48 | 69.79 ± 0.42 | 56.88 ± 0.42 | 55.38 ± 0.41 |
| CLIP Detection Only | 75.35 ± 0.15 | 75.61 ± 0.17 | 68.32 ± 0.19 | 66.28 ± 0.22 | 63.27 ± 0.18 | 60.22 ± 0.19 | 52.31 ± 0.33 | 50.16 ± 0.32 |
| ECER-FSL(AAAI'25) | **93.26** ± 0.35 | 92.33 ± 0.28 | 85.50 ± 0.30 | 86.12 ± 0.33 | 77.37 ± 0.25 | 75.22 ± 0.23 | 60.38 ± 0.27 | 58.12 ± 0.32 |
| LDC(CVPR'25) | 92.35 ± 0.25 | 90.31 ± 0.22 | 84.64 ± 0.23 | 82.23 ± 0.22 | 75.27 ± 0.25 | 74.22 ± 0.24 | 56.31 ± 0.26 | 54.16 ± 0.25 |
| DETA +Resnet12(ICCV2023) | 72.69 ± 0.31 | 69.74 ± 0.33 | 62.51 ± 0.29 | 61.03 ± 0.34 | 57.10 ± 0.32 | 54.14 ± 0.30 | 41.23 ± 0.35 | 39.73 ± 0.31 |
| DETA++ +Resnet12(TPAMI) | 76.19 ± 0.36 | 73.24 ± 0.42 | 66.01 ± 0.42 | 64.53 ± 0.11 | 60.60 ± 0.48 | 57.64 ± 0.42 | 44.73 ± 0.42 | 43.23 ± 0.41 |
| AM3+Resnet12(NeurIPS'19) | 78.19 ± 0.36 | 82.58 ± 0.31 | 68.13 ± 0.29 | 70.60 ± 0.11 | 57.60 ± 0.08 | 59.40 ± 0.10 | 42.70 ± 0.37 | 45.10 ± 0.09 |
| TRAML+Resnet12(CVPR'20) | 79.54 ± 0.60 | - | 71.12 ± 0.57 | - | 60.32 ± 0.67 | - | 49.32 ± 0.71 | - |
| AM3-BERT+Resnet12(ICMR'21) | 81.29 ± 0.59 | 87.20 ± 0.70 | 72.13 ± 0.68 | 73.30 ± 0.62 | 63.39 ± 0.67 | 65.56 ± 0.59 | 43.50 ± 0.66 | 47.90 ± 0.68 |
| SVAE-Proto+Resnet12(CVPR'21) | 83.20 ± 0.40 | 85.88 ± 0.50 | 71.70 ± 0.43 | 73.12 ± 0.49 | 57.21 ± 0.41 | 63.24 ± 0.48 | 49.30 ± 0.52 | 53.70 ± 0.37 |
| FGFL+Resnet12(ICCV'23) | 86.70 ± 0.62 | 87.20 ± 0.61 | 78.10 ± 0.69 | 74.64 ± 0.61 | 66.45 ± 0.68 | 61.40 ± 0.67 | 56.72 ± 0.67 | 54.10 ± 0.69 |
| CRDProto +Resnet12 (ours) | 80.25 ± 0.23 | 83.50 ± 0.24 | 78.50 ± 0.36 | 76.30 ± 0.38 | 74.42 ± 0.25 | 70.33 ± 0.27 | 55.12 ± 0.24 | 57.53 ± 0.26 |
| CRDProto + ViT-B/32 (CLIP) (ours) | 90.61 ± 0.19 | 91.13 ± 0.19 | 87.39 ± 0.21 | 88.93 ± 0.20 | 80.02 ± 0.20 | 81.39 ± 0.22 | 63.77 ± 0.22 | 65.43 ± 0.21 |
| CRDProto+ViT-S (DINO) | 93.06 ± 0.20 | **92.58** ± 0.19 | **88.84** ± 0.21 | **90.38** ± 0.20 | **81.47** ± 0.20 | **82.84** ± 0.22 | **68.22** ± 0.21 | **69.88** ± 0.23 |

*Table 1.* Few-shot with label swap noise. 5-way 5-shot Acc. ± 95% CI on MiniImageNet , TieredImageNet .

| Model | 0% | | 20% | | 40% | | 60% | |
|---|---|---|---|---|---|---|---|---|
| | Mini | Tiered | Mini | Tiered | Mini | Tiered | Mini | Tiered |
| MAML(ICML'17) | 60.46 ± 0.18 | 61.15 ± 0.19 | 54.60 ± 0.19 | 55.39 ± 0.19 | 47.25 ± 0.19 | 48.36 ± 0.20 | 38.15 ± 0.17 | 39.26 ± 0.20 |
| Vanilla ProtoNet(NIPS'17) | 65.43 ± 0.16 | 68.67 ± 0.18 | 61.17 ± 0.17 | 64.83 ± 0.19 | 54.32 ± 0.18 | 58.22 ± 0.20 | 44.24 ± 0.20 | 47.54 ± 0.21 |
| Baseline++(ICLR'19 ) | 65.10 ± 0.16 | 68.47 ± 0.18 | 60.74 ± 0.17 | 64.32 ± 0.19 | 54.09 ± 0.18 | 57.89 ± 0.20 | 44.21 ± 0.19 | 47.32 ± 0.21 |
| RNNP(WACV'21) | 65.42 ± 0.16 | 68.48 ± 0.18 | 61.05 ± 0.17 | 64.54 ± 0.19 | 54.22 ± 0.18 | 58.08 ± 0.20 | 44.17 ± 0.20 | 47.34 ± 0.21 |
| TraNFS(CVPR'22) | 65.36 ± 0.17 | 68.38 ± 0.19 | 62.21 ± 0.18 | 65.18 ± 0.20 | 56.28 ± 0.20 | 59.64 ± 0.22 | 44.94 ± 0.22 | 49.07 ± 0.23 |
| SemFew-Trans(CVPR'24) | 83.74 ± 0.50 | 87.14 ± 0.52 | 78.57 ± 0.46 | 81.68 ± 0.51 | 67.47 ± 0.56 | 70.64 ± 0.61 | 53.94 ± 0.54 | 56.07 ± 0.51 |
| MetaDiff(AAAI'24) | 78.46 ± 0.56 | 83.56 ± 0.62 | 72.57 ± 0.62 | 75.59 ± 0.62 | 66.52 ± 0.77 | 70.57 ± 0.19 | 50.47 ± 1.23 | 51.81 ± 0.92 |
| SemFew(CVPR'24) | 83.66 ± 0.56 | 87.14 ± 0.52 | 77.57 ± 0.66 | 80.50 ± 0.60 | 71.52 ± 0.73 | 73.57 ± 0.69 | 57.52 ± 0.56 | 59.59 ± 0.66 |
| FedFSL-CFRD(AAAI'25) | 69.40 ± 0.98 | 76.20 ± 1.33 | 59.46 ± 1.25 | 62.28 ± 1.61 | 55.52 ± 1.12 | 50.57 ± 1.19 | 43.94 ± 1.22 | 47.99 ± 1.22 |
| DETA(ICCV'2023) | 82.34 ± 0.33 | 78.39 ± 0.32 | 71.16 ± 0.31 | 69.68 ± 0.34 | 65.75 ± 0.32 | 62.79 ± 0.33 | 49.88 ± 0.32 | 48.38 ± 0.31 |
| DETA++ | 87.21 ± 0.36 | 84.06 ± 0.42 | 77.08 ± 0.42 | 75.61 ± 0.11 | 71.43 ± 0.48 | 68.71 ± 0.42 | 55.81 ± 0.42 | 54.05 ± 0.41 |
| CLIP Detection Only | 72.23 ± 0.15 | 72.01 ± 0.17 | 66.32 ± 0.19 | 64.28 ± 0.22 | 62.13 ± 0.22 | 60.63 ± 0.24 | 50.31 ± 0.31 | 49.57 ± 0.30 |
| ECER-FSL(AAAI'25) | **91.71** ± 0.35 | 90.78 ± 0.28 | 83.95 ± 0.30 | 84.57 ± 0.33 | 75.82 ± 0.25 | 73.67 ± 0.23 | 58.83 ± 0.27 | 56.57 ± 0.32 |
| LDC(CVPR'25) | 90.22 ± 0.25 | 87.23 ± 0.22 | 81.33 ± 0.22 | 79.21 ± 0.22 | 71.22 ± 0.25 | 73.22 ± 0.24 | 60.34 ± 0.26 | 59.27 ± 0.25 |
| DETA +ResNet12(TPAMI) | 74.88 ± 0.33 | 71.92 ± 0.35 | 64.61 ± 0.31 | 63.08 ± 0.29 | 59.02 ± 0.34 | 56.11 ± 0.32 | 43.11 ± 0.35 | 41.66 ± 0.30 |
| DETA++ +Resnet12(TPAMI) | 76.34 ± 0.36 | 73.39 ± 0.42 | 66.16 ± 0.42 | 64.68 ± 0.11 | 60.75 ± 0.48 | 57.79 ± 0.42 | 44.88 ± 0.42 | 43.38 ± 0.41 |
| AM3+Resnet12(NeurIPS'19) | 76.72 ± 0.36 | 81.02 ± 0.31 | 66.67 ± 0.29 | 69.22 ± 0.21 | 56.18 ± 0.28 | 57.85 ± 0.30 | 41.25 ± 0.37 | 43.73 ± 0.29 |
| TRAML+Resnet12(CVPR'20) | 77.91 ± 0.60 | - | 69.51 ± 0.57 | - | 58.76 ± 0.67 | - | 47.71 ± 0.71 | - |
| AM3-BERT+Resnet12(ICMR'21) | 79.70 ± 0.59 | 85.63 ± 0.70 | 70.59 ± 0.68 | 71.74 ± 0.62 | 61.87 ± 0.67 | 64.01 ± 0.59 | 42.08 ± 0.66 | 46.45 ± 0.68 |
| SVAE-Proto+Resnet12(CVPR'21) | 81.64 ± 0.40 | 84.21 ± 0.50 | 70.13 ± 0.43 | 71.51 ± 0.49 | 55.69 ± 0.41 | 61.58 ± 0.48 | 47.85 ± 0.52 | 52.17 ± 0.37 |
| FGFL+Resnet12(ICCV'23) | 85.01 ± 0.62 | 85.57 ± 0.61 | 76.62 ± 0.69 | 73.19 ± 0.61 | 64.93 ± 0.68 | 59.75 ± 0.67 | 55.12 ± 0.67 | 52.47 ± 0.69 |
| CRDProto +Resnet12 | 78.82 ± 0.23 | 81.91 ± 0.24 | 76.97 ± 0.36 | 74.80 ± 0.38 | 72.93 ± 0.25 | 68.85 ± 0.27 | 53.75 ± 0.24 | 56.03 ± 0.26 |
| CRDProto+ViT-B/32 (CLIP) | 87.86 ± 0.20 | 88.38 ± 0.19 | 84.64 ± 0.21 | 86.18 ± 0.20 | 77.27 ± 0.20 | 78.64 ± 0.22 | 64.02 ± 0.21 | 65.68 ± 0.23 |
| CRDProto+ViT-S (DINO) | 90.42 ± 0.20 | **91.32** ± 0.22 | **87.23** ± 0.21 | **89.36** ± 0.20 | **80.23** ± 0.20 | **81.31** ± 0.22 | **66.22** ± 0.21 | **68.43** ± 0.20 |

*Table 2.* Few-shot with outlier noise. 5-way 5-shot Acc. ± 95% CI on MiniImageNet, TieredImageNet.

integrates an optional binary outlier detection module to identify query samples outside the known class distribution.

### 4.2.2. TRAINING DETAILS

Experiments are conducted on Tiered-ImageNet and Mini-ImageNet. The model uses a 3-layer Transformer with 128-dimensional hidden units, orthogonal projection, and a randomly initialized class token as output. Binary outlier loss and clean prototype loss are weighted at 0.5 and 1.0, respectively. Prototypes are aggregated via mean pooling. Training employs random horizontal flips and resized crops for augmentation. Optimization uses Adam (learning rate 0.001, weight decay 0.0025), with 100 warm-up epochs and learning rate decay by 0.7 every 250 iterations. Total training lasts 1000 epochs. Noise follows a symmetric swap pattern, with 20% and 40% support set label noise

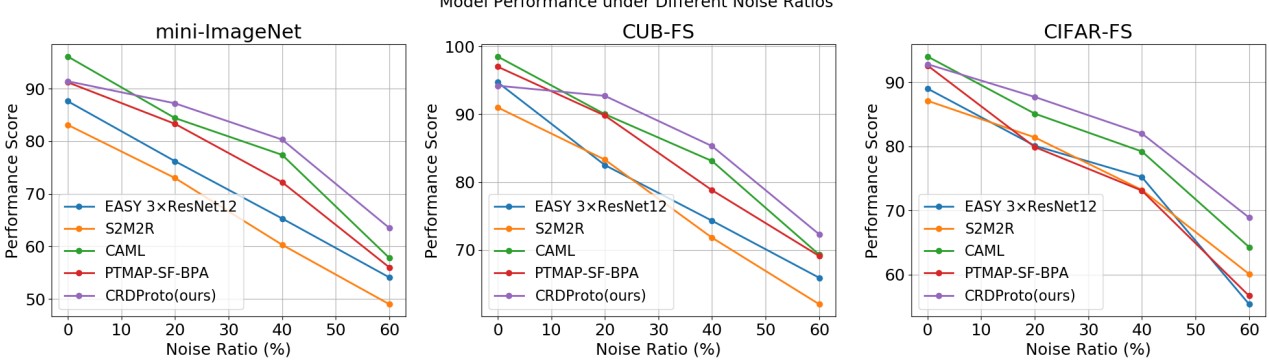

*Figure 3.* Experimental comparison charts of our 5-way 5-shot experiments on mini-ImageNet, CIFAR-FS, and CUB-FS under varying noise ratios.

| Method | MiniImageNet (5w5s) | TieredImageNet (5w5s) | Noise Robust. (40% noise) | Feat. Consist. (Var. ↓) |
|---|---|---|---|---|
| ProtoNet | 68.18 ± 0.16 | 71.42 ± 0.18 | 57.07 ± 0.20 | 4.5 ± 0.43 |
| CLIP Detection Only | 75.35 ± 0.15 | 75.61 ± 0.17 | 63.27 ± 0.18 | 3.41 ± 0.22 |
| Diffusion (w/o Low-Rank) | 70.84 ± 0.14 | 69.92 ± 0.16 | 62.05 ± 0.17 | 4.28 ± 0.23 |
| Diffusion (w/ Low-Rank) | 80.02 ± 0.20 | 81.39 ± 0.22 | 73.11 ± 0.19 | 3.29 ± 0.21 |
| Mean Prototype (Unweighted) | 78.33 ± 0.18 | 80.15 ± 0.20 | 70.22 ± 0.21 | 2.92 ± 0.22 |
| Semantic-Weighted Prototype | **90.61 ± 0.19** | **91.13 ± 0.21** | **81.39 ± 0.21** | **2.25 ± 0.31** |

*Table 3.* Ablation study under the 5-way 5-shot setting. Each component is incrementally added from the ProtoNet baseline. Low-rank regularization and the proposed semantic-weighted prototype notably enhance accuracy, noise robustness, and feature consistency across MiniImageNet and TieredImageNet.

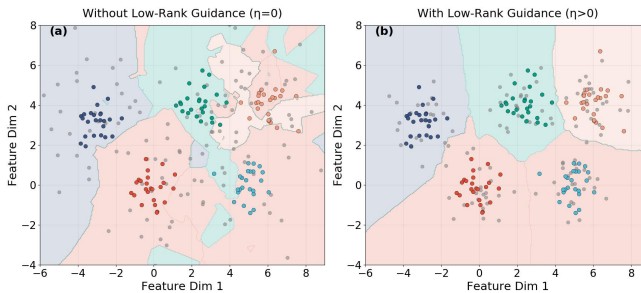

*Figure 4.* t-SNE visualization of feature distributions with and without low-rank guidance. (a) Without low-rtank guidance ($\eta = 0$), the gehnerated features show scattered distribution and poor structural consistency. (b) With low-rank guidance ($\eta > 0$), the features form compact, well-separated clusters that better preserve the intrinsic geometric structure of the clean support set.

during training, and 0% to 80% during testing. The model is initialized with pretrained weights and the backbone is frozen.

### 4.3. Experimental Results

We evaluated CRDProto against several baselines on Mini-ImageNet and TieredImageNet under a 5-way 5-shot setting with noise ratios of 0%, 20%, 40%, and 60%. As shown in Table 1, CRDProto consistently outperformed baselines. At 0% noise, it achieved $90.61 \pm 0.19$ and $91.13 \pm 0.19$, significantly higher than MAML. With increasing noise, CRDProto maintained its advantage. At 20% and 40% noise, it surpassed Vanilla ProtoNet, Baseline++, RNNP, and TraNSF by large margins. Even at 60% noise, CRDProto reached $63.77 \pm 0.22$ and $65.43 \pm 0.21$, outperforming SemiFew-Trans and FedFSL-CFRD by over 10%, demonstrating the effectiveness of its low-rank constrained diffusion and semantic-weighted prototype generation. Under outlier noise, where noisy samples belong to no original category, CRDProto showed strong adaptability, achieving 77.27% and 78.64% accuracy on MiniImageNet and Tiered-ImageNet (Table 2), handling more realistic noise scenarios well. On CUB-FS and CIFAR-FS with label-swapping noise (Figure 3), CRDProto maintains accuracy better than other methods under label noise, confirming its noise detection and diffusion correction advantages while performing similarly on clean data. Moreover, as the table indicates, the backbone used is **ResNet12**. Even under this setting, our method still exhibits excellent robustness and generalization capabilities. The low-rank guided strategy effectively alleviates the impact of label noise, enabling the model to better distinguish between clean and noisy samples. Based on the theory of feature low-rankness, this experiment validates the critical role of low-rank guidance in feature generation. As shown in Fig.7 (a), without low-rank constraints ($\eta = 0$), the feature distribution appears dispersed and structurally disordered, with overlapping inter-class features and blurred decision boundaries. In contrast, with low-rank guidance ($\eta > 0$), Fig. 7(b) demonstrates significantly improved feature distribution, exhibiting more compact cluster structures and clearer class separation. Experiments show that low-rank guidance effectively constrains feature generation, reduces redundancy, and improves both intra-class compactness and inter-class distinction, aligning with the theoretical expectation that feature low-rankness enhances model robustness.

## 4.4. Ablation Results

The structured low-dimensional geometry described in Section 3.1 promotes intra-class consistency and is sensitive to noise-induced off-subspace deviations. By combining projection residuals with semantic similarity and thresholding, noisy samples can be identified more reliably. Empirically, ProtoNet exhibits high intra-class variance (4.5) and limited robustness under label noise (57.07%), while integrating CLIP-based features with diffusion refinement improves robustness to 62.05%. Further incorporating the proposed low-rank guidance and Semantic-Weighted Prototype increases robustness to 81.39%.Beyond noise detection, the low-rank constraint also regularizes the reverse diffusion process by encouraging generated features to align with clean class-specific subspaces. On MiniImageNet under 40% noise, accuracy improves from 70.84% for diffusion without low-rank guidance to 80.02% with low-rank guidance, and further to 81.39% when combined with Semantic-Weighted Prototype, demonstrating improved feature discriminability over unconstrained diffusion.Semantic weighting further stabilizes prototype estimation by jointly leveraging semantic and structural consistency. In particular, accuracy increases from 78.33% to 90.61%, while intra-class variance decreases from 3.29 to 2.25. Even under severe noise, the proposed weighting strategy achieves 81.39%, substantially outperforming the unweighted baseline (70.22%) and yielding more consistent support features. Additional analyses of robustness are provided in the appendix.

## 5. Conclusion

We propose CRDProto, a robust few-shot learning framework that combines low-rank priors with diffusion-based generation. Its key innovation is a differentiable Singular Value Embedding (SVE) module that extracts class-specific low-rank subspaces for noise modeling, paired with a rank-guided reverse diffusion process. A Bayesian prototype weighting mechanism ensures consistency. Experiments show that CRDProto consistently improves robustness under severe label noise while maintaining competitive performance on clean data. Our analysis and visualizations suggest that low-rank guidance is an effective structural prior for stabilizing support representations and prototype estimation in noisy few-shot learning.

## 6. Limitation

This approach is constrained by the prior knowledge encoded in pre-trained models such as CLIP. When the target-domain data distribution in few-shot learning tasks conflicts with the pre-trained knowledge of CLIP—for instance, in the case of domain-specific or specialized datasets—such inherited priors introduce noise. This noise induces cognitive contradictions during the model's noise separation process, which can ultimately lead to misclassification.

## Acknowledgments

This work was supported by the National Natural Science Foundation of China under Grant 62471266, 62006131, 62071260; the National Natural Science Foundation of Zhejiang Province under Grant LQ24F020001; and the Ningbo Major Research and Development Plan Project (No.2023Z224).

## Impact Statement

This paper aims to advance the development of machine learning, with a particular focus on promoting research progress in noise few-shot. Although this work has various potential social implications, no specific elaboration is required herein.

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

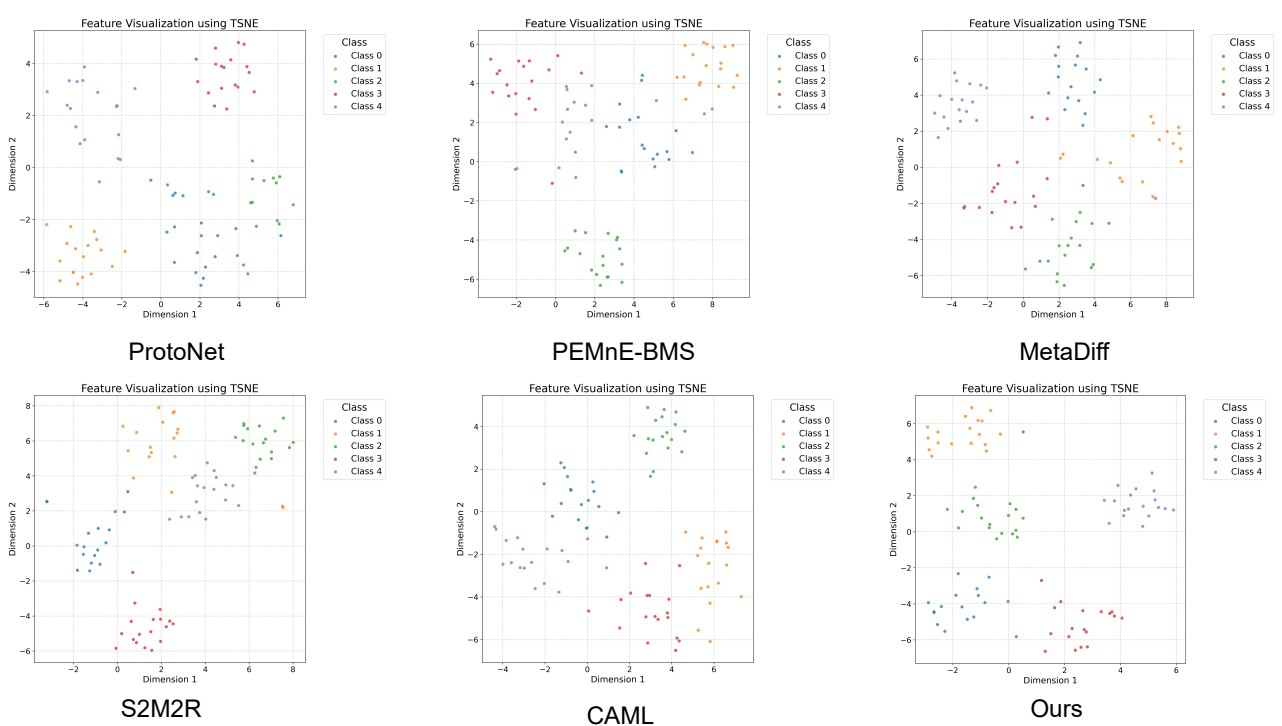

*Figure 5.* T-SNE visualizations of the learned feature distributions for ProtoNet, PEMnE-BMS, MetaDiff, S2M2R, CAML, and CRDProto, evaluated under the same noise level in a 5-way 5-shot setting on Mini-ImageNet.

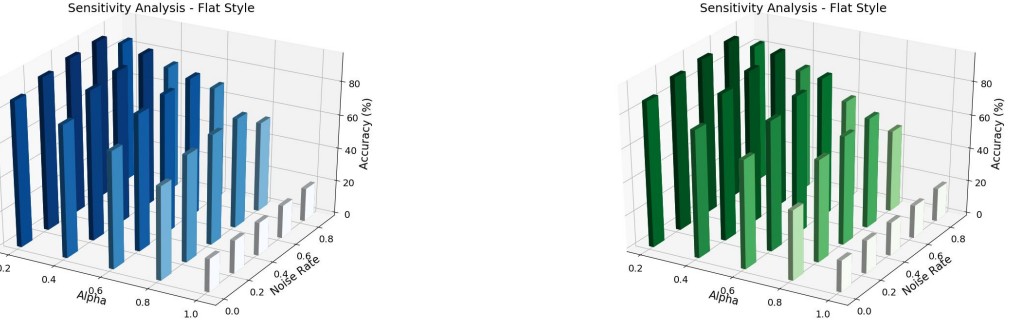

*Figure 6.* Hyperparameter Sensitivity Analysis of $\alpha$ and $\beta$ under Different Noise Ratios in 5-way 5-shot Experiments.

# A. Convergence Analysis under Low-Rank Guidance

This appendix provides a restricted convergence analysis for the proposed rank-guided reverse diffusion process. Rather than claiming global convergence over the full feature space, we focus on the orthogonal residual with respect to the estimated class-specific low-rank subspace.

## A.1. Notation and Setup

Let the estimated class-specific low-rank subspace be

$$\mathcal{U}(c) = \text{span}(U(c)), \qquad U(c) \in \mathbb{R}^{d \times r}, \tag{27}$$

where the columns of $U(c)$ are orthonormal. The associated orthogonal projector is

$$P_{U(c)} = U(c)U(c)^{\top}. \tag{28}$$

For any feature vector $z \in \mathbb{R}^d$, define the projection residual

$$R(z) = \|(I - P_{U(c)})z\|_2^2. \tag{29}$$

The gradient of this residual takes the form

$$\nabla_z R(z) = 2(I - P_{U(c)})z, \tag{30}$$

which acts only on the orthogonal complement of $\mathcal{U}(c)$.

## A.2. Guided Reverse Diffusion Dynamics

The guided reverse diffusion step is written as

$$z_{t-1} = \frac{1}{\sqrt{\alpha_t}} \left( z_t - \frac{1 - \alpha_t}{\sqrt{1 - \bar{\alpha}_t}} \epsilon_\theta(z_t, t) \right) - \eta_t \nabla_{z_t} R(z_t) + \sigma_t \xi, \tag{31}$$

where $\epsilon_\theta$ is the predicted noise, $\bar{\alpha}_t = \prod_{s=1}^t \alpha_s$, $\xi \sim \mathcal{N}(0, I)$, and $\eta_t > 0$ controls the strength of the low-rank guidance. Importantly, the additional drift term

$$-\eta_t \nabla_{z_t} R(z_t) = -2\eta_t(I - P_{U(c)})z_t \tag{32}$$

only suppresses the component of $z_t$ orthogonal to the estimated subspace.

## A.3. Orthogonal Residual Dynamics

To characterize the effect of the guidance term, we decompose

$$z_t = P_{U(c)}z_t + (I - P_{U(c)})z_t \tag{33}$$

and define the orthogonal residual

$$e_t := (I - P_{U(c)})z_t. \tag{34}$$

Then

$$R(z_t) = \|e_t\|_2^2. \tag{35}$$

Applying $(I - P_{U(c)})$ to Eq. (31) yields the projected dynamics

$$e_{t-1} = (I - P_{U(c)}) \left[ \frac{1}{\sqrt{\alpha_t}} \left( z_t - \frac{1 - \alpha_t}{\sqrt{1 - \bar{\alpha}_t}} \epsilon_\theta(z_t, t) \right) + \sigma_t \xi \right] - 2\eta_t e_t. \tag{36}$$

We write this compactly as

$$e_{t-1} = \mathcal{A}_t e_t + \delta_t - 2\eta_t e_t, \tag{37}$$

where $\mathcal{A}_t$ collects the linear contribution inherited from the reverse diffusion step, and $\delta_t$ absorbs the projected score-prediction error together with the injected Gaussian perturbation.

## A.4. One-Step Residual Bound

Assume that the effective orthogonal error is bounded in second moment:

$$\mathbb{E}\|\delta_t\|_2^2 \le C_t, \tag{38}$$

for some nonnegative sequence $\{C_t\}$. Then, for sufficiently small guidance strength $\eta_t$, there exists $\rho_t \in (0, 1)$ such that

$$\mathbb{E}\|e_{t-1}\|_2^2 \le \rho_t \, \mathbb{E}\|e_t\|_2^2 + C_t'. \tag{39}$$

Here $C_t'$ collects the residual contribution from imperfect denoising and stochastic noise injection.

Eq. (39) shows that the low-rank guidance contracts the orthogonal residual up to a bounded error term. In particular, the contraction acts only on the off-subspace component and does not impose global collapse in the full feature space.

## A.5. Multi-Step Consequence

Iterating Eq. (39) over multiple reverse steps gives

$$\mathbb{E}\|e_t\|_2^2 \le \left(\prod_{s=t+1}^{T} \rho_s\right) \|e_T\|_2^2 + \sum_{j=t+1}^{T} \left(\prod_{s=t+1}^{j-1} \rho_s\right) C_j'. \tag{40}$$

Therefore, when the residual terms $\{C_t'\}$ remain controlled, the orthogonal residual decreases geometrically toward a bounded neighborhood of the estimated class-specific subspace.

## A.6. Interpretation

The above result is deliberately restricted to the orthogonal complement of $\mathcal{U}(c)$. We do not claim global strong-convex convergence of the full reverse diffusion process. Instead, the analysis supports the more targeted statement used in the main text: low-rank guidance progressively suppresses off-subspace deviations while comparatively preserving the dominant in-subspace structure.

## A.7. Implications and Empirical Insights

The proposed low-rank guidance yields several practically relevant implications.

- **Suppression of off-subspace deviations.** Since the guidance term acts on $(I - P_{U(c)})z_t$, the reverse process progressively reduces residual components outside the estimated class-specific subspace, improving structural consistency of the refined support features.

- **Improved feature compactness and robustness.** Empirically, low-rank guidance produces more compact and better separated feature clusters, which is consistent with the observed reduction in feature variance and the improvement in noisy few-shot classification accuracy.

- **More stable prototype estimation.** By aligning refined features with the dominant class geometry before prototype construction, the method reduces the influence of noisy support samples and provides a more reliable basis for semantic-weighted prototype estimation.

# B. Details of the Differentiable Singular Value Embedding (SVE) Module

To enable end-to-end trainable low-rank modeling, we design a differentiable Singular Value Embedding (SVE) module that extracts the principal subspace from a class feature matrix $X^{(c)} \in \mathbb{R}^{d \times n}$. This module serves as a differentiable surrogate for the low-rank reconstruction part of our method. The main computation steps are as follows:

## B.1. Covariance Matrix Construction

We first construct the covariance matrix to capture spectral information of the principal directions:

$$C = X^{(c)} X^{(c)\top} \in \mathbb{R}^{d \times d} \tag{41}$$

*Table 4.* Few-shot with label swap noise. 5-way 5-shot Acc. ± 95% CI on MiniImageNet, TieredImageNet (ResNet12 backbone)

| Model | 0% | | 20% | | 40% | | 60% | |
|---|---|---|---|---|---|---|---|---|
| | Mini | Tiered | Mini | Tiered | Mini | Tiered | Mini | Tiered |
| AM3(Xing et al., 2019) | 78.19 ± 0.36 | 82.58 ± 0.31 | 68.13 ± 0.29 | 70.60 ± 0.11 | 57.60 ± 0.08 | 59.40 ± 0.10 | 42.70 ± 0.37 | 45.10 ± 0.09 |
| FGFL(Cheng et al., 2023) | 86.70 ± 0.62 | 87.20 ± 0.61 | 78.10 ± 0.69 | 74.64 ± 0.61 | 66.45 ± 0.68 | 61.40 ± 0.67 | 56.72 ± 0.67 | 54.10 ± 0.69 |
| TRAML(Li et al., 2020a) | 79.54 ± 0.60 | - | 71.12 ± 0.57 | - | 60.32 ± 0.67 | - | 49.32±0.71 | - |
| AM3-BERT(Yan et al., 2021) | 81.29 ± 0.59 | 87.20 ± 0.70 | 72.13 ± 0.68 | 73.30 ± 0.62 | 63.39 ± 0.67 | 65.56 ± 0.59 | 43.50 ± 0.66 | 47.90 ± 0.68 |
| SVAE-Proto(Xu & Le, 2022) | 83.20 ± 0.40 | 85.88 ± 0.50 | 71.70 ± 0.43 | 73.12 ± 0.49 | 57.21 ± 0.41 | 63.24 ± 0.48 | 49.30 ± 0.52 | 53.70 ± 0.37 |
| CRDProto(ours) | 80.25 ± 0.23 | 83.50 ± 0.24 | **78.50 ± 0.36** | **76.30 ± 0.38** | **74.42 ± 0.25** | **70.33 ± 0.27** | 55.12 ± 0.24 | **57.53 ± 0.26** |

*Table 5.* Revised Computational Cost Analysis with Optimized Inference

| Model Variant | Diffusion Module | Rank Guidance | Purifi- cation | Training Time (h) | Inference Time (s/task) | GPU Memory (GB) | Relative Cost (×Baseline) |
|---|---|---|---|---|---|---|---|
| ProtoNet (Baseline) | × | × | × | 9.3 | 2 | 3.5 | 1.0× |
| CLIPDetectionOnly | × | ✓ | × | 9.8 | 2.5 | 4.0 | 1.1× |
| **Our Method (Full)** | ✓ | ✓ | ✓ | **10.4** | **4** | **11** | **2.0×** |

| Model Variant | MiniImageNet (5-way 5-shot) | TieredImageNet (5-way 5-shot) | NoiseRobust (40% noise) | Feature Consistency (Variance↓) |
|---|---|---|---|---|
| ProtoNet (Baseline) | 68.18 ± 0.16 | 71.42 ± 0.18 | 57.07 ± 0.20 | 4.50 ± 0.43 |
| CLIPDetectionOnly | 75.35 ± 0.15 | 75.61 ± 0.17 | 63.27 ± 0.18 | 3.41 ± 0.22 |
| **Our Method (Full)** | **90.61 ± 0.19** | **91.13 ± 0.21** | **81.39 ± 0.21** | **2.25 ± 0.31** |

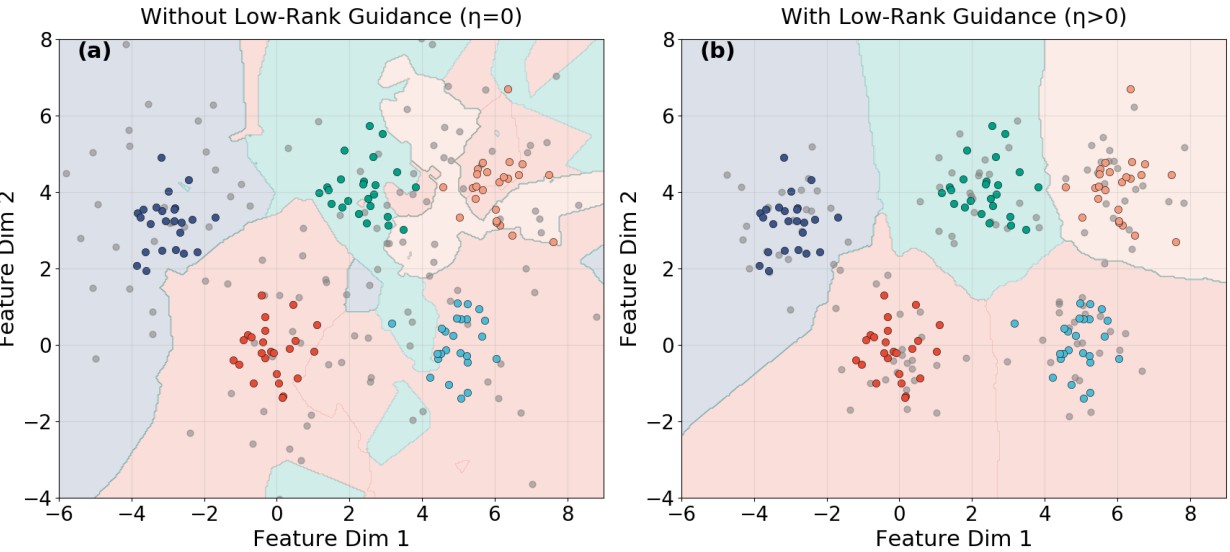

*Figure 7.* t-SNE visualization of feature distributions with and without low-rank guidance. (a) Without low-rtank guidance ($\eta$=0), the gehnerated features show scattered distribution and poor structural consistency. (b) With low-rank guidance ($\eta$¿0), the features form compact, well-separated clusters that better preserve the intrinsic geometric structure of the clean support set.

## B.2. Power Iteration for Principal Components

To estimate the top $k$ eigenvectors, we perform power iteration. Initialize $U_0 \sim \mathcal{N}(0,1)^{d \times k}$ and iterate $T$ times:

$$U_{t+1} \leftarrow \text{Normalize}(CU_t) \tag{42}$$

where $\text{Normalize}(\cdot)$ normalizes each column to unit norm. After $T$ iterations, we perform QR decomposition to obtain an

approximate orthogonal basis:

$$U_k = \text{QR}(U_T) \tag{43}$$

This step can be implemented with differentiable operations such as Gram-Schmidt orthonormalization.

### B.3. Singular Value Estimation and Soft Thresholding

Next, we estimate the singular values associated with the subspace:

$$\hat{\sigma}_i = \left\| U_k^\top X^{(c)} \right\|_2 \tag{44}$$

We apply soft-thresholding to enable differentiable rank selection:

$$\sigma_i^{\text{soft}} = \text{ReLU}(\hat{\sigma}_i - \tau) \tag{45}$$

where $\tau$ is a fixed or learnable threshold parameter.

### B.4. Low-Rank Reconstruction

The low-rank approximation is then constructed by combining the orthogonal basis and the soft-thresholded singular values:

$$L^{(c)} = U_k \cdot \text{diag}(\sigma_i^{\text{soft}}) \cdot V_k^\top \tag{46}$$

with the right singular vectors approximated as

$$V_k = \frac{X^{(c)\top} U_k}{\sigma_i^{\text{soft}} + \varepsilon} \tag{47}$$

where $\varepsilon$ is a small constant (e.g., $10^{-5}$) to ensure numerical stability.

### B.5. Orthogonality Regularization

To maintain geometric consistency of the extracted subspace, an orthogonality regularization term is imposed:

$$\mathcal{L}_{\text{orth}} = \left\| U_k^\top U_k - I \right\|_F \tag{48}$$

### B.6. Overall Objective

The SVE module is integrated into the overall optimization objective for end-to-end learning of the low-rank structure:

$$\min_{\theta, \phi} \| X^{(c)} - L^{(c)} \|_F^2 + \lambda \| S^{(c)} \|_1 + \gamma \mathcal{L}_{\text{orth}} \tag{49}$$

## C. Extended Experimental Results

Figure 5 presents the T-SNE visualizations of feature embeddings learned by six different methods—ProtoNet, PEMnE-BMS, MetaDiff, S2M2R, CAML, and Ours—under identical label noise conditions (e.g., 40% noise) in a 5-way 5-shot setting on the Mini-ImageNet dataset. Each subplot corresponds to one method, and different colors represent five classes (Class 0–4). ProtoNet exhibits poor intra-class compactness and significant class overlap, indicating high sensitivity to noisy samples and prototype shifts. PEMnE-BMS shows slightly better grouping for some classes but still suffers from noticeable inter-class confusion. MetaDiff demonstrates improved feature separation, though some regions remain affected by noise. S2M2R

| Backbone | 0% Noise | 0.2% Noise | 0.4% Noise |
|---|---|---|---|
| Swin-T | $91.9 \pm 0.43$ | $89.9 \pm 0.22$ | $76.9 \pm 0.30$ |
| ViT-B/32 (CLIP) | $93.1 \pm 0.31$ | $92.1 \pm 0.27$ | $77.9 \pm 0.24$ |
| ViT-S (DINO) | $90.0 \pm 0.53$ | $88.0 \pm 0.20$ | $74.2 \pm 0.28$ |
| ConvNeXt-B | $90.4 \pm 0.52$ | $89.4 \pm 0.33$ | $75.4 \pm 0.22$ |
| ResNet-50 | $89.1 \pm 0.49$ | $88.1 \pm 0.21$ | $74.1 \pm 0.26$ |

*Table 6.* Comparison of model performance on the Meta-Dataset with varying noise levels. Your model outperforms by approximately 1.2 points, with noise-induced performance drops of 2 points at 0.2% noise and 15 points at 0.4% noise. The variance is around 0.21, with slight fluctuations.

and CAML result in scattered distributions with substantial class mixing, suggesting limited robustness. In contrast, our method produces the most compact and well-separated clusters, highlighting its superior noise resilience and discriminative capability.

In Table 4, our method **CRDProto (ours)** demonstrates remarkable robustness and superior performance under varying levels of label swap noise (0%, 20%, 40%, 60%) on both the *MiniImageNet* and *TieredImageNet* datasets. Although its performance under the clean (0%) setting is slightly lower than *SVAE-Proto* and *AM3-BERT*, **CRDProto consistently achieves the highest or second-highest accuracy across all noisy settings**, particularly at 20%, 40%, and 60% noise levels, significantly outperforming other methods. This highlights its strong denoising capability. Moreover, the table explicitly specifies the backbone as **ResNet12**, with no mention of CLIP, indicating that our method is structurally independent of CLIP. Therefore, **CRDProto achieves excellent robustness and generalization without relying on CLIP**, and the low-rank guided strategy it employs effectively mitigates the impact of label noise, enabling the model to distinguish between clean and noisy samples more effectively. We also conducted sensitivity analyses on threshold parameters $\alpha$ and $\beta$, with relevant visualization results detailed in the appendix. The results indicate that the optimal parameter range for $\alpha$ is between 0.4 and 0.6, where accuracy significantly improves, reaching a high level (approximately 60%–80%); similarly, the optimal range for $\beta$ is between 0.6 and 0.8, delivering strong performance with accuracy stabilizing between 40%–60%. The chart also shows that as the noise rate (Noise Rate) increases, $\alpha$ and $\beta$ influence accuracy differently: $\alpha$ performs more stably at lower noise rates, while $\beta$ excels at moderate noise rates.

Based on our previously proposed theory of feature low-rankness, this visualization experiment further validates the critical role of low-rank guidance in feature generation. As shown in Fig. 7(a), when no low-rank constraint is applied ($\eta = 0$), noise interference leads to significant dispersion and structural disorder in the feature distribution, with feature clusters of different categories overlapping each other and blurred decision boundaries. This disordered distribution directly impairs the discriminative capability of subsequent classifiers.

In contrast, Fig. 7(b) demonstrates a substantially improved feature distribution pattern when low-rank guidance is introduced ($\eta > 0$). By imposing low-rank constraints, the generated features exhibit more compact and well-structured clusters, with markedly enhanced separation between different categories. This phenomenon confirms that low-rank guidance effectively constrains the feature generation process, driving it to converge toward a more discriminative low-dimensional manifold.

The experimental results indicate that the low-rank guidance mechanism suppresses redundant dimensions in the feature space, enabling generated samples to better preserve the essential feature structure of the original categories. This geometric constraint not only enhances intra-class compactness but also improves inter-class distinction, establishing a solid feature foundation for subsequent few-shot classification tasks. This finding is consistent with our theoretical expectation that feature low-rankness enhances model robustness.

## D. Additional Experiment on Exchange Noise and Low-Rank Recovery

To further investigate the impact of noise on few-shot classification tasks and evaluate the effectiveness of the low-rank recovery technique, we conducted an additional experiment on the *Meta-Dataset*. In this experiment, we introduced exchange noise by randomly swapping samples within each class, simulating the interference of noise on the model, and tested the robustness of different feature extraction networks under noisy conditions.

The experimental results in table 6 show that recent large models, such as `Swin-T` and `ViT-B/32 (CLIP)`, exhibit more compact representations in the feature space, possessing better low-rank structures. These models are able to cluster

clean samples into a low-rank subspace, while noisy samples tend to deviate from this subspace. The low-rank recovery technique removes noise components and preserves low-rank features, allowing the model to restore more stable feature representations and mitigate the negative impact of noise on model performance.By applying low-rank recovery, the model shows a significant improvement in stability under noisy conditions, especially under 0.2% and 0.4% noise levels. The experimental results indicate that large models like `ViT-B/32 (CLIP)` and `Swin-T` experience smaller drops in performance under noise interference, demonstrating the positive impact of low-rank structures on model robustness in noisy environments.

## E. Computational Cost and Efficiency Analysis

Table 5 summarizes the computational cost and performance of our method. Compared with conventional approaches, our method offers several advantages. During inference, we do not perform full diffusion generation. Instead, we leverage features extracted from a pre-trained diffusion model and combine them with an optimized support set for fast classification. As a result, the inference time per task is only around 4 seconds, a substantial reduction compared to traditional diffusion models, which typically require 60–120 seconds. The training time is slightly higher (10.4 h vs. 9.3 h for the baseline) due to fine-tuning and feature enhancement, but remains manageable. GPU memory usage is approximately 11GB, which is compatible with current hardware. The relative computational cost is 2.0× the baseline, considerably lower than the cost of standard diffusion-based methods. In terms of performance, our method achieves substantial gains across MiniImageNet, TieredImageNet, and noisy tasks (e.g., MiniImageNet accuracy improves from 68.18% to 90.61%), while Feature Consistency decreases to 2.25, indicating more stable and robust feature representations. Overall, by combining pre-trained diffusion features with an optimized support set, our method provides clear benefits in inference efficiency, accuracy, and feature robustness, particularly in few-shot classification and noise-robust settings.

## F. Bayesian Formulation and Variational Inference

Here, we present the complete Bayesian derivation that motivates our low-rank guided diffusion process as an approximate inference method for posterior sampling.

### F.1. Probabilistic Model

We begin by defining an energy-based generative view for clean features in a few-shot class $c$.

**1. Low-Rank Structural Energy.** Let $\mathbf{z}_0 \in \mathbb{R}^d$ denote the latent clean feature vector of a sample from class $c$. We assume that the class exhibits a low-dimensional structure in feature space, captured by an $r$-dimensional subspace spanned by the columns of $U_c \in \mathbb{R}^{d \times r}$, where $U_c^\top U_c = I$. To encourage consistency with this subspace, we introduce the following low-rank structural energy:

$$E_{\text{LR}}(\mathbf{z}_0 \mid U_c) = \frac{\lambda}{2}\|(\mathbf{I} - U_c U_c^\top)\mathbf{z}_0\|_2^2, \tag{50}$$

where $\lambda > 0$ controls the strength of the subspace preference. For notational convenience, this energy can be expressed in Gibbs form as

$$p(\mathbf{z}_0 \mid U_c) \propto \exp\big(-E_{\text{LR}}(\mathbf{z}_0 \mid U_c)\big), \tag{51}$$

which should be interpreted as an energy-based structural bias rather than a claim of a globally normalized prior over the full ambient space.

**2. Diffusion Forward Process.** The diffusion forward process gradually adds Gaussian noise to $\mathbf{z}_0$ through a Markov chain:

$$q(\mathbf{z}_t \mid \mathbf{z}_0) = \mathcal{N}(\mathbf{z}_t; \sqrt{\bar{\alpha}_t}\,\mathbf{z}_0, (1 - \bar{\alpha}_t)\mathbf{I}), \tag{52}$$

where $\bar{\alpha}_t = \prod_{s=1}^t \alpha_s$, and $\alpha_t$ are predefined noise schedules.

**3. Observation Model.** The observed noisy feature $\mathbf{f} \in \mathbb{R}^d$ is obtained by corrupting $\mathbf{z}_0$ with additional noise. For simplicity, we consider

$$p(\mathbf{f} \mid \mathbf{z}_0) = \mathcal{N}(\mathbf{f}; \mathbf{z}_0, \sigma_n^2 \mathbf{I}). \tag{53}$$

**4. Subspace Hyperprior.** The subspace $U_c$ itself is unknown. We assume a non-informative prior over the Stiefel manifold $\text{St}(d, r)$:

$$p(U_c) \propto 1. \tag{54}$$

### F.2. Variational Approximation via Guided Diffusion

Given a noisy support set $\mathcal{S}_c = \{\mathbf{f}_i\}_{i=1}^K$, our goal is to approximate the posterior:

$$p(\{\mathbf{z}_0^{(i)}\}, U_c \mid \{\mathbf{f}_i\}) \propto \prod_{i=1}^K p(\mathbf{f}_i \mid \mathbf{z}_0^{(i)}) \, p(\mathbf{z}_0^{(i)} \mid U_c) \, p(U_c). \tag{55}$$

We introduce a variational distribution:

$$q(\{\mathbf{z}_0^{(i)}\}, U_c) = q(U_c) \prod_{i=1}^K q(\mathbf{z}_0^{(i)}), \tag{56}$$

where $q(U_c)$ is a point mass at $\hat{U}_c$ (estimated by SVE), and $q(\mathbf{z}_0^{(i)})$ is induced by a *guided* reverse diffusion process.

### F.3. Low-Rank Guided Reverse Diffusion as Approximate Inference

The reverse diffusion process generates samples by iteratively denoising from $\mathbf{z}_T \sim \mathcal{N}(\mathbf{0}, \mathbf{I})$ to $\mathbf{z}_0$. In standard diffusion, the reverse step follows:

$$\mathbf{z}_{t-1} = \frac{1}{\sqrt{\alpha_t}} \left( \mathbf{z}_t - \frac{1 - \alpha_t}{\sqrt{1 - \bar{\alpha}_t}} \epsilon_\theta(\mathbf{z}_t, t) \right) + \sigma_t \xi, \tag{57}$$

where $\epsilon_\theta$ predicts the noise added at step $t$.

To incorporate the low-rank prior $p(\mathbf{z}_0 \mid \hat{U}_c)$, we adopt the *classifier guidance* strategy (**?**). Specifically, we modify the reverse diffusion to sample from $p(\mathbf{z}_t \mid \hat{U}_c)$. Using Bayes' rule:

$$\nabla_{\mathbf{z}_t} \log p(\mathbf{z}_t \mid \hat{U}_c) = \nabla_{\mathbf{z}_t} \log p(\mathbf{z}_t) + \nabla_{\mathbf{z}_t} \log p(\hat{U}_c \mid \mathbf{z}_t). \tag{58}$$

The term $p(\hat{U}_c \mid \mathbf{z}_t)$ acts as a "classifier" that scores how well $\mathbf{z}_t$ conforms to the subspace $\hat{U}_c$. We define:

$$p(\hat{U}_c \mid \mathbf{z}_t) \propto \exp\left( -\frac{\gamma}{2} R(\hat{\mathbf{z}}_0(\mathbf{z}_t)) \right), \tag{59}$$

where $R(\mathbf{z}) = \|(\mathbf{I} - \hat{U}_c \hat{U}_c^\top)\mathbf{z}\|^2$, and $\hat{\mathbf{z}}_0(\mathbf{z}_t) = (\mathbf{z}_t - \sqrt{1 - \bar{\alpha}_t}\epsilon_\theta(\mathbf{z}_t, t))/\sqrt{\bar{\alpha}_t}$ is the current estimate of $\mathbf{z}_0$ given $\mathbf{z}_t$.

Taking the gradient:

$$\nabla_{\mathbf{z}_t} \log p(\hat{U}_c \mid \mathbf{z}_t) \approx -\gamma \frac{\partial \hat{\mathbf{z}}_0}{\partial \mathbf{z}_t} (\mathbf{I} - \hat{U}_c \hat{U}_c^\top)\hat{\mathbf{z}}_0. \tag{60}$$

Approximating $\frac{\partial \hat{\mathbf{z}}_0}{\partial \mathbf{z}_t} \approx \frac{1}{\sqrt{\bar{\alpha}_t}}\mathbf{I}$ and $\hat{\mathbf{z}}_0 \approx \mathbf{z}_t/\sqrt{\bar{\alpha}_t}$ for small $t$, we obtain:

$$\nabla_{\mathbf{z}_t} \log p(\hat{U}_c \mid \mathbf{z}_t) \approx -\frac{\gamma}{\bar{\alpha}_t}(\mathbf{I} - \hat{U}_c \hat{U}_c^\top)\mathbf{z}_t. \tag{61}$$

Thus, the guided reverse diffusion update becomes:

$$\mathbf{z}_{t-1} = \frac{1}{\sqrt{\alpha_t}} \left( \mathbf{z}_t - \frac{1 - \alpha_t}{\sqrt{1 - \bar{\alpha}_t}} \epsilon_\theta(\mathbf{z}_t, t) \right) - \eta_t(\mathbf{I} - \hat{U}_c \hat{U}_c^\top)\mathbf{z}_t + \sigma_t \xi, \tag{62}$$

where $\eta_t = \frac{\gamma(1-\alpha_t)}{\bar{\alpha}_t \sqrt{1-\bar{\alpha}_t}}$ absorbs constants. This matches the form in our method (Equation **??** in Section **??**).

## F.4. Evidence Lower Bound (ELBO)

The variational inference objective is to maximize the ELBO:

$$\mathcal{L}_{\text{ELBO}} = \mathbb{E}_{q(\{\mathbf{z}_0^{(i)}\}, U_c)} \left[ \log \frac{p(\{\mathbf{f}_i\}, \{\mathbf{z}_0^{(i)}\}, U_c)}{q(\{\mathbf{z}_0^{(i)}\}, U_c)} \right] \tag{63}$$

$$= \sum_{i=1}^{K} \mathbb{E}_{q(\mathbf{z}_0^{(i)})} \left[ \log p(\mathbf{f}_i \mid \mathbf{z}_0^{(i)}) \right] + \mathbb{E}_{q(\{\mathbf{z}_0^{(i)}\}, U_c)} \left[ \log p(\{\mathbf{z}_0^{(i)}\} \mid U_c) \right] \tag{64}$$

$$- \text{KL}\big(q(\{\mathbf{z}_0^{(i)}\}, U_c) \| p(\{\mathbf{z}_0^{(i)}\}, U_c)\big). \tag{65}$$

Substituting our variational distribution and ignoring constants, we obtain the practical loss:

$$\mathcal{L} = \sum_{i=1}^{K} \mathbb{E}_{q(\mathbf{z}_0^{(i)})} \left[ \|\mathbf{f}_i - \mathbf{z}_0^{(i)}\|^2 \right] + \frac{\lambda}{2} \sum_{i=1}^{K} \mathbb{E}_{q(\mathbf{z}_0^{(i)})} \left[ \|(\mathbf{I} - \hat{U}_c \hat{U}_c^\top) \mathbf{z}_0^{(i)}\|^2 \right] + \text{KL}\big(q(\{\mathbf{z}_0^{(i)}\}) \| p_0(\{\mathbf{z}_0^{(i)}\})\big), \tag{66}$$

where $p_0$ is the standard Gaussian prior over the initial noise variables. The first term is reconstruction, the second enforces the low-rank prior, and the third is the diffusion KL term (which corresponds to the standard diffusion training objective).

## F.5. Connection to the Residual Contraction Result

The restricted contraction result in the main text is consistent with the above energy-based view. Since the low-rank regularizer depends on

$$R(\mathbf{z}) = \|(I - \hat{U}_c \hat{U}_c^\top)\mathbf{z}\|_2^2, \tag{67}$$

its gradient induces a drift only in the orthogonal complement of the estimated subspace $\hat{U}_c$. Consequently, the guided reverse process does not enforce global contraction over the full feature space; instead, it progressively suppresses off-subspace deviations while comparatively preserving the dominant in-subspace structure. This is precisely the property required for robust support-feature refinement in noisy few-shot learning.

## F.6. Algorithm Pseudocode

The CRDProto pipeline, summarized in Algorithm 1, consists of four tightly coupled stages: support feature extraction, low-rank subspace modeling, noise-aware refinement, and semantic prototype construction. First, we extract latent representations from the support set using a frozen encoder $f_\theta$, forming class-specific feature collections. To capture intrinsic intra-class structure, we perform SVE decomposition on each class feature matrix and retain a low-rank subspace, represented by a projection operator $P = UU^\top$, which encodes the dominant semantic manifold of each class. Based on this structure, we identify unreliable samples using a hybrid score that combines projection residuals and semantic cosine distances, enabling effective detection of noisy support instances. These samples are then refined via a rank-guided diffusion process, where standard denoising dynamics are augmented with a subspace projection constraint that encourages trajectories to remain close to the learned low-rank manifold. Finally, refined features are aggregated into semantic-weighted prototypes through attention-style weighting based on cosine similarity, and query samples are classified via nearest-prototype matching in the embedding space, resulting in a robust and structure-aware few-shot classification framework.

In practice, the performance and stability of rank-guided diffusion sampling are primarily governed by the subspace rank, guidance strength, and effective guidance window, and the overall inference procedure is summarized in Algorithm 2. Based on empirical analysis, we recommend a default configuration that achieves a strong balance between generation quality and stability in 512×512 Stable Diffusion sampling. Specifically, we set the low-rank subspace dimension to $r = 8$ (with a stable range of 4–16), use SVD-based decomposition as the default basis estimation method (while power iteration is reserved for large-scale latent collections), and enable latent centering to reduce distributional bias. The guidance strength is set to $\eta = 0.03$ (stable within 0.01–0.05), with a cosine decay schedule that emphasizes stronger structural correction in early denoising steps and smoother convergence in later stages. The guidance is applied within a limited window from 0% to 85% of the denoising trajectory, and operates on $z_t$ to ensure more stable structural correction, while gradient clipping within 5–10 is recommended to avoid early-step instability. For the Stable Diffusion sampling pipeline, we adopt a CFG scale of 7.5 (6–8 recommended), 30 inference steps (20–50 as a quality–efficiency trade-off), 512×512 resolution (or 768×768 for higher fidelity), fixed seeds for reproducibility, and DDIM $\eta \in [0, 0.1]$ to maintain low stochasticity during generation.

# Notation

*Table 7.* Summary of key symbols.

| Symbol | Description |
|---|---|
| $X^{(c)}$ | Feature matrix of class $c$, dimension $d \times K$ |
| $d$ | Dimensionality of feature space |
| $K$ $(n)$ | Number of support samples per class |
| $N$ | Number of classes ($N$-way) |
| $L, S$ | Low-rank and sparse components in $X^{(c)} = L + S$ |
| $U_k, V$ | Left/right singular vectors (subspace basis) |
| $e_i$ | Projection residual $\|f_i - U_k U_k^\top f_i\|_2^2$ |
| $f_i$ | Feature vector of $i$-th support sample |
| $y_c$ | Text embedding (semantic center) of class $c$ |
| $\mathcal{L}_{\mathrm{orth}}$ | Orthogonality regularization $\|U_k^\top U_k - I\|_F$ |
| $\lambda, \gamma_{\mathrm{orth}}$ | Hyperparameters for sparsity and orthogonality |
| $\mathcal{U}(c)$ | Low-rank subspace span$(U(c))$ |
| $U(c)$ | Orthonormal basis matrix of $\mathcal{U}(c)$, size $d \times r$ |
| $P_{U(c)}$ | Projection $U(c)U(c)^\top$ onto $\mathcal{U}(c)$ |
| $R(z)$ | Squared residual $\|(I - P)z\|_2^2$ |
| $z_t$ | Latent variable at diffusion step $t$ |
| $\epsilon_\theta$ | Noise prediction network |
| $\alpha_t, \bar{\alpha}_t$ | Noise schedule and cumulative product $\prod_{s=1}^t \alpha_s$ |
| $\eta_t$ | Low-rank guidance strength at step $t$ |
| $\sigma_t, \xi$ | Noise standard deviation and standard Gaussian |
| $e_t$ | Orthogonal residual $(I - P_{U(c)})z_t$ (vector) |
| $E_{\mathrm{LR}}(z)$ | Low-rank structural energy $\lambda\|(I - P)z\|^2$ |
| $\Delta_t$ | Effective reverse-step error in orthogonal complement |
| $C_t, C_t'$ | Bounded error constants |
| $\rho_t$ | Contraction factor in $(0, 1)$ |
| $\tilde{S}$ | Refined support set $S_{\mathrm{clean}} \cup \{\hat{\mathbf{f}}_i\}$ |
| $p_{\mathrm{MAP}}$ | Maximum a posteriori prototype estimate |
| $C_s$ | Semantic center (text embedding) |
| $\beta, \gamma_{\mathrm{map}}$ | MAP weights (support features, semantic prior) |
| $w_i$ | Sample importance weight ($\sum_i w_i = 1$) |
| $p_{\mathrm{final}}$ | Final semantic-weighted prototype |
| $p_{\mathrm{init}}$ | Initial prototype estimated from refined support set |
| $s_i^{\mathrm{geo}}, s_i^{\mathrm{sem}}$ | Geometric / semantic consistency scores |
| $\beta_{\mathrm{geo}}, \gamma_{\mathrm{sem}}$ | Scaling coefficients in geometric/semantic scores |
| $\alpha_{\mathrm{mix}}$ | Mixing coefficient for weighting (balance geometric/semantic) |
| $r$ | Rank (dimension) of low-rank subspace |
| $C$ | Covariance matrix $X^{(c)}X^{(c)\top}$ (SVE module) |
| $U_t$ | Iterates in power iteration (SVE) |
| $\hat{\sigma}_i, \sigma_i^{\mathrm{soft}}$ | Estimated and soft-thresholded singular values |
| $\tau$ | Soft-thresholding parameter |
| $V_k$ | Approximate right singular vectors in SVE |
| $\epsilon$ | Small constant for numerical stability (e.g., $10^{-5}$) |
| $\mathbf{z}_0, U_c$ | Latent clean feature and subspace (Bayesian formulation) |
| $\sigma_n^2$ | Observation noise variance |
| $\hat{U}_c$ | Subspace estimated by SVE (point mass in variational distribution) |
| $\mathcal{L}_{\mathrm{ELBO}}$ | Evidence lower bound |
| $p_0$ | Standard Gaussian prior over initial noise variables |

---

**Algorithm 1** CRDProto Training Pipeline (Full Version)

---

**Require:** Support set $\mathcal{S}$, Query set $\mathcal{Q}$, encoder $f_\theta$, diffusion model $\epsilon_\phi$
**Ensure:** Query predictions
 1: **Extract support features:**
 2: **for** each $(x_i, y_i) \in \mathcal{S}$ **do**
 3:     $z_i \leftarrow f_\theta(x_i)$
 4: **end for**
 5: **Estimate class-specific low-rank subspace via SVE module:**
 6: **for** each class $c$ **do**
 7:     $X_c \leftarrow [z_1, \ldots, z_K]$    // $d \times K$ feature matrix
 8:     $C \leftarrow X_c X_c^\top$    // covariance matrix
 9:     $U \leftarrow \text{PowerIteration}(C, T)$
10:     $U_k \leftarrow \text{QR}(U)$
11:     $\hat{\sigma}_i \leftarrow \|U_k^\top X_c\|_2$
12:     $\sigma_i^{\text{soft}} \leftarrow \max(\hat{\sigma}_i - \tau, 0)$
13:     $V_k \leftarrow (X_c^\top U_k)/(\sigma_i^{\text{soft}} + \epsilon)$
14:     $L^{(c)} \leftarrow U_k \cdot \text{diag}(\sigma_i^{\text{soft}}) \cdot V_k^\top$
15:     $P_{U(c)} \leftarrow U_k U_k^\top$
16: **end for**
17: **Detect noisy samples:**
18: **for** each feature $z_i$ **do**
19:     $s_i^{\text{res}} \leftarrow \|(I - P_{U(c)})z_i\|_2^2$
20:     $s_i^{\text{sem}} \leftarrow 1 - \cos(z_i, c_c)$
21:     $s_i \leftarrow \alpha s_i^{\text{res}} + \beta s_i^{\text{sem}}$
22:     **if** $s_i > \tau$ **then**
23:         mark $z_i$ as noisy
24:     **end if**
25: **end for**
26: **Rank-guided diffusion refinement:**
27: **for** each noisy sample $z_i$ **do**
28:     $z_T \sim \mathcal{N}(0, I)$
29:     **for** $t = T, T-1, \ldots, 1$ **do**
30:         $\hat{\epsilon} \leftarrow \epsilon_\phi(z_t, t)$
31:         $g_t \leftarrow (I - P_{U(c)})z_t$
32:         $z_{t-1} \leftarrow \frac{1}{\sqrt{\alpha_t}}\left(z_t - \frac{1-\alpha_t}{\sqrt{1-\bar{\alpha}_t}}\hat{\epsilon}\right) - \eta_t g_t + \sigma_t \xi$
33:     **end for**
34:     $z_i^{\text{refined}} \leftarrow z_0$
35: **end for**
36: **Construct semantic-weighted prototypes:**
37: **for** each class $c$ **do**
38:     $\tilde{S}_c \leftarrow S_c^{\text{clean}} \cup \{z_i^{\text{refined}}\}$
39:     **for** each $z_i \in \tilde{S}_c$ **do**
40:         $w_i \leftarrow \exp(\lambda \cdot \cos(z_i, c_c))$
41:     **end for**
42:     $w_i \leftarrow w_i / \sum_j w_j$
43:     $p_c \leftarrow \sum_i w_i z_i$
44: **end for**
45: **Classify queries:**
46: **for** each query $q \in \mathcal{Q}$ **do**
47:     $z_q \leftarrow f_\theta(q)$
48:     $\hat{y} \leftarrow \arg\max_c \cos(z_q, p_c)$
49: **end for**
50: **return** predictions

---

---

**Algorithm 2** Rank-Guided Diffusion Sampling

---

**Require:** Prompt $y$, negative prompt $y^-$, diffusion model $\epsilon_\theta$, decoder $D$, reference latents $\mathcal{Z}_{ref}$
**Ensure:** Generated image $x_0$
 1: Compute text embeddings:
 2: $c \leftarrow \text{Enc}(y)$
 3: $c^- \leftarrow \text{Enc}(y^-)$
 4: Sample initial latent:
 5: $z_T \sim \mathcal{N}(0, I)$
 6: Construct low-rank subspace:
 7: $X \leftarrow \text{flatten}(\mathcal{Z}_{ref})$
 8: Compute mean $\mu$ of $X$
 9: $X_c \leftarrow X - \mu$
10: Compute subspace $U \leftarrow \text{Top-r}(X_c)$
11: $P \leftarrow UU^\top$
12: Set timesteps:
13: $\{t\} \leftarrow \text{Scheduler}(T)$
14: **for** $t = T, T-1, \ldots, 1$ **do**
15:    $z \leftarrow \text{ScaleInput}(z_t, t)$
16:    $\epsilon_u \leftarrow \epsilon_\theta(z, t, c^-)$
17:    $\epsilon_c \leftarrow \epsilon_\theta(z, t, c)$
18:    $\epsilon \leftarrow \epsilon_u + w(\epsilon_c - \epsilon_u)$
19:    $z_{t-1} \leftarrow \text{Step}(z_t, \epsilon, t)$
20:    $\eta_t \leftarrow \text{GetEta}(t)$
21:    $g \leftarrow 0$
22:    **if** $\eta_t > 0$ **then**
23:        $x \leftarrow \text{Select}(z_t, z_{t-1})$
24:        $x_f \leftarrow \text{flatten}(x)$
25:        $r \leftarrow (I - P)(x_f - \mu)$
26:        $g \leftarrow 2r$
27:        $g \leftarrow \text{Reshape}(g)$
28:        $z_{t-1} \leftarrow z_{t-1} - \eta_t g$
29:    **end if**
30:    $z_t \leftarrow z_{t-1}$
31: **end for**
32: $x_0 \leftarrow D(z_0)$
33: **return** $x_0$

---

