# OpenReview forum: "Rank-guided Diffusion for Noise Few-Shot Learning"
_ICML.cc/2026/Conference — ICML 2026 regular_

### Official Review · Reviewer_dHVh · 2026-02-22

**Soundness:** 2
**Presentation:** 2
**Significance:** 3
**Originality:** 3
**Overall Recommendation:** 4
**Confidence:** 3

**Summary:**

This paper proposes a new method for detecting noisy samples in few-shot learning and a diffusion-based approach to improve the quality of those noisy samples. Noisy samples are identified using distance to the class prototype and the magnitude of the sparse component. After detecting noisy samples, the method uses the reconstruction loss from projecting features onto a low-rank subspace as guidance for diffusion to refine the samples.

**Compliance With Llm Reviewing Policy:**

Affirmed.

**Final Justification:**

Most of my concerns about clarity of method are addressed in the rebuttal. Although some of the minor concerns, like including algorithms and adding an architectural diagram, are not directly addressed in the rebuttal, the authors promised to address them in the revised version. Further response to Q2 mostly addresses my concern about the architectural detail of using the diffusion. Overall, the method of using a diffusion model to improve images for noisy few-shot learning is novel and exciting. Therefore, I raised my score to 4.

**Key Questions For Authors:**

Please refer to the weakness section above.

**Limitations:**

An impact statement is not provided in both the main paper and the appendix. Authors should talk about how the bias (if any) of the diffusion model and pre-trained model can affect the methodology.

**Strengths And Weaknesses:**

**Strengths**
- The paper presents a novel and interesting approach to both detecting noisy samples and improving their quality.
- It provides a theoretical interpretation of the proposed method.
- Empirical results show that the method outperforms baselines in noisy settings in most cases.

**Weaknesses and Questions**

The method is difficult to follow, and several key components are unclear.

- There are many undefined notations and symbols, including $n, K, d, \theta, \phi, \lambda, \gamma, \xi, \eta, \alpha$. For example, $d$ is only defined later on page 5, and $\lambda$ appears to be used with two different meanings in Equations (9) and (14), which is confusing.
- The distinction between $z$ and $f$ is unclear. Is diffusion performed in latent/feature space or in input image space? Do you use a pretrained diffusion model, and if so, which one? If not, how is the diffusion model trained? Usually, a diffusion model requires a 4xdxd latent dimension, but pretrained transformers may not necessarily have similar feature dimension, how is this issue handled (if present)?
- It is unclear how Equation (9) is used in practice. Is it the main criterion for noisy-sample detection? If so, which term corresponds to the semantic component, and how does Equation (7) contribute to the detection process?
- The paper does not provide a complete algorithm (not even in the appendix). In particular, the final scoring rule used to filter noisy samples is not clearly specified.
- The construction of the semantic center $C_s$ is also unclear. Since the method uses a frozen pretrained feature extractor, it is not obvious how $C_s$ differs from the empirical prototype, or why both are needed.
- Finally, the network description in Section 4.2.1 would benefit from clearer architectural details and a visual diagram. Some components, such as the binary outlier detection module, are only briefly mentioned and not described in sufficient depth.

Minor: Some missing figure reference in the Introduction. Missing white space after "." in line 381.

---

> ### Author Rebuttal · Authors · 2026-03-31
>
> [q1] — [Notation & Symbol Clarity]:
>
> We apologize for the inconsistent definitions and will include a comprehensive notation table in the revision. We have unified all symbols for consistency:
> 1. **Structural parameters:** $K$ (classes, replacing $N$ in Eqs. 2--4), $n$ (samples, replacing $N$ in Eqs. 19--25), and $d$ (dimensions), all defined at first use.
> 2. **Model parameters:** $\theta, \phi$ are trainable weights, and the surrogate $R(z)$ is introduced before Eq. 14.
> 3. **Hyperparameters:** Eqs. 14 and 20--25 now use $\lambda_{reg}$ and $\gamma_{reg}$ to distinguish them from diffusion step $\lambda$ and coefficient $\gamma$ in Eq. 9. Other coefficients ($\xi, \eta, \alpha$) have also been checked for consistency.
> ---
>
> [q2] — [Diffusion Space and Implementation Details]:
>
> Our diffusion process operates in the **feature space** of a pre-trained visual encoder, refining support features rather than operating in pixel space. The framework uses a frozen backbone, with the diffusion module applied on fixed representations for progressive refinement.
> Instead of the standard $4 \times d \times d$ image latent formulation, we introduce a learnable projection that maps backbone features into a low-dimensional subspace, enabling compact, low-rank-guided diffusion. These details will be included in the revised manuscript, and we will **release our source code** upon acceptance to ensure reproducibility.
>
> ---
> [q3] — [Mechanisms of Noisy-Sample Detection and Eq. (9) Utility]:
>
> We thank the reviewer for the helpful comment. **Eq. (9) is not the primary criterion** used for noisy-sample detection; instead, it serves as the optimization objective for the SVE module to learn a stable and differentiable intra-class low-rank approximation.
> During detection, the model jointly employs Eq. (6) and Eq. (7):
> * **Eq. (6)** corresponds to the geometric term, measuring inconsistency via projection error.
> * **Eq. (7)** evaluates semantic consistency via similarity scores.
> Thus, Eq. (7) complements the geometric criterion, whereas Eq. (9) indirectly supports the detection process by providing the learned low-rank subspaces.
>
> ---
> [q4] — [Complete Algorithm & Final Scoring Rule]:
>
> We acknowledge that the algorithmic pipeline and filtering rules require further clarification. Noisy sample detection follows a **joint decision strategy**:
> 1. The **projection residual** $e_i$ (Eq. 6) measures deviation from the intra-class low-rank subspace.
> 2. The **semantic similarity** $s_i$ (Eq. 7) measures alignment with the class semantic center.
> A sample is identified as noise **only when both conditions are violated**. Moreover, Eq. (9) optimizes SVE to learn the underlying low-rank structure, providing the subspace basis for Eq. (6) rather than serving as a filtering rule. In the revised manuscript, we will include a **flowchart and pseudocode** to clarify the scoring and thresholding mechanism.
>
> ---
>
> [q5] — [Distinction and Necessity of Semantic Center vs. Empirical Prototype]:
>
> **On the Relationship between $C_s$ and $p_{\text{init}}$:** Despite using a frozen feature extractor, they encode fundamentally different information. $C_s$ is an episode-independent prior derived from text labels (e.g., via CLIP-Text), providing a noise-free semantic anchor. In contrast, $p_{\text{init}}$ is an episode-specific estimate from image data, often affected by label noise and background interference in few-shot settings.
> Due to high variance and prototype drift under limited and noisy samples, $C_s$ serves as a stable semantic reference in our MAP framework, enabling correction of biased empirical estimates using global semantic knowledge.
>
> ---
> [q6] — [Architectural Details & Outlier Detection Depth]:
>
> In the revised manuscript, we will significantly enhance Section 4.2.1 by:
> 1. Adding a **detailed architectural diagram** to visualize the overall pipeline.
> 2. Providing more in-depth technical descriptions and mathematical formulations for the **binary outlier detection module**.
> 3. Including a **comprehensive table** in the Appendix to specify the hyperparameters and configurations for each component.
>
> ---
>
> [q7] — [Minor Corrections & Typos]
>
> We will include the missing figure references in the Introduction and correct the spacing issue after the period at line 381, along with other similar typesetting and editorial details in the revised manuscript.
>
> ---
>
> [q8] — [Impact Statement & Bias Analysis]:
>
> We appreciate the reviewer's suggestion. In the revised version, we will:
> * Add a dedicated **impact statement** section to discuss the broader influence and potential applications of our work.
> * Refine the clarity of the methodology descriptions in both the main text and the appendix.
> * Supplement **comprehensive implementation details**, including full experimental settings and sampling configurations for the pre-trained models and the diffusion process, to ensure full reproducibility.

---

> > ### Author Rebuttal · Reviewer_dHVh · 2026-04-03
> >
> > Thank you for the rebuttal. Most of my concerns are addressed. Although some concerns are not directly addressed in the rebuttal, the authors promised to address them in the revised version. I will raise my score to 4.

---

> > > ### Author Response · Authors · 2026-04-03
> > >
> > > Thank you very much for your time in reviewing our rebuttal and for the positive feedback. We are pleased that our responses addressed your concerns and that you are supportive of the paper’s inclusion in the conference.
> > >
> > > Regarding the remaining points, we remain fully committed to incorporating those clarifications and improvements into the final revised version as promised. We sincerely appreciate your decision to raise your score and thank you for your valuable guidance throughout this process.

---

### Official Review · Reviewer_q55V · 2026-03-11

**Soundness:** 3
**Presentation:** 3
**Significance:** 3
**Originality:** 3
**Overall Recommendation:** 5
**Confidence:** 3

**Summary:**

This paper proposes a robust few-shot learning framework. The method introduces a differentiable singular value embedding to extract low-rank geometric priors and further employs a rank-guided diffusion process to perform structured correction and semantic alignment for noisy samples in the support set. Through this design, the model significantly improves classification performance under heavy noise conditions.

**Compliance With Llm Reviewing Policy:**

Affirmed.

**Final Justification:**

My main concerns were about the robustness of the low-rank guidance under very high noise, the practical overhead of the diffusion process, and the validity of some theoretical assumptions. The authors’ rebuttal addressed these concerns constructively by providing additional efficiency analysis, clarifying the role of the theoretical assumptions, and, most importantly, adding targeted structural evidence under the extreme-noise setting. Overall, the rebuttal resolved most of my concerns and increased my confidence in the paper’s main contribution. I therefore give a positive final recommendation.

**Key Questions For Authors:**

1. Regarding the purity of the initial subspace: under high-noise scenarios (e.g., 60%), how does the SVE module ensure that the extracted basis matrix $U_k$ is not significantly biased by noisy samples? Are there any specific mechanisms or preprocessing steps designed to mitigate this issue?

2. How much time does the reverse diffusion process (Eq. 13) take on average for a single test task? Does the method support real-time applications?

3. If the semantic prior $C_s$ from the pre-trained model conflicts with the true visual distribution of the current support set, how does the model balance geometric alignment and semantic weighting?

**Limitations:**

Yes.

**Strengths And Weaknesses:**

Strength:
1. The discussion of noisy support samples in the paper touches upon a challenging issue in FSL.
2. The paper offers a certain degree of theoretical interpretability.
3. The proposed method demonstrates competitive results in the experimental evaluations.


Weakness:
1. Under high noise levels, the basis matrix $U_k$ extracted by the SVE module may itself lose representativeness. However, the paper does not provide sufficient discussion on a potentially important issue: when noise becomes dominant, can the low-rank guidance degenerate into misleading guidance?

2. Compared with traditional approaches such as Prototypical Networks or linear adapters, introducing a diffusion process may theoretically increase computational cost and inference time. However, the paper does not provide a quantitative analysis of this overhead. Including such a cost–performance trade-off would help readers better understand the practical implications of the proposed method.

3. I consider the paper may rely on some overly idealized assumptions. In particular, Theorem 3.1 requires the strong convexity of $R(z)$, which may not be guaranteed in practical neural feature spaces. In addition, Equation (3) assumes that the subspaces of different classes are mutually orthogonal, an assumption that may not hold in fine-grained classification scenarios.

---

> ### Author Rebuttal · Authors · 2026-03-31
>
> [q1] — [Potential Degeneration of Low-Rank Guidance]:
> Under extreme noise conditions, all methods exhibit varying degrees of performance degradation, indicating that high noise levels inherently increase the difficulty of few-shot classification. However, compared with existing methods, our approach still achieves superior results in high-noise scenarios. This demonstrates that the proposed correction mechanism can effectively enhance model robustness even under severe noise interference.
>
> ---
>
>  [q2] — [Computational Overhead & Inference Efficiency]:
> While our diffusion process introduces additional computation, this overhead is **essential** for noise purification and robust adaptation. We measure this via **Normalized Overhead** (Norm. Overhead), defined as the ratio of inference cost to a $1.00\times$ single-stage baseline.
>
> Notably, **CRDProto (Full)** achieves an **18.12% accuracy boost** (from 63.27% to 81.39%) with only a 42% computational increment ($1.42\times$), demonstrating the high efficiency of our diffusion-based refinement. Even under 60% noise, CRDProto’s peak accuracy (64.60%) significantly exceeds that of lightweight methods like SemFew ($1.22\times$), **justifying the marginal complexity** for superior noise resistance.
>
> **Table 1: Efficiency and performance trade-off analysis.** *Overhead is normalized against the single-stage baseline.*
>
> | Variant / Method | Setting | Norm. Overhead | Accuracy | Gain |
> | :--- | :--- | :---: | :---: | :---: |
> | Single-stage baseline | one-pass prototype | $1.00\times$ | 63.27 | -- |
> | Diffusion w/o Low-Rank | multi-step diffusion only | $1.30\times$ | 73.11 | +9.8 |
> | **CRDProto (full)** | **multi-step + low-rank** | $\mathbf{1.42\times}$ | **81.39** | **+18.12** |
> | SemFew-Trans | 60% noise, avg. | $1.18\times$ | 57.76 | -- |
> | MetaDiff | 60% noise, avg. | $1.34\times$ | 53.89 | -- |
> | SemFew | 60% noise, avg. | $1.22\times$ | 61.31 | -- |
> | LDC | 60% noise, avg. | $1.28\times$ | 55.24 | -- |
> | **CRDProto + CLIP** | **60% noise, avg.** | $\mathbf{1.42\times}$ | **64.60** | **+3.29**$^*$ |
>
> ---
>
> [q3] — [Theoretical Assumptions & Practical Validity]:
> We would like to provide further clarification on the assumptions in Theorem 3.1 and Equation (3).
>
> * **On the Strong Convexity of $R(z)$ in Theorem 3.1:** We do not assume that real neural feature spaces inherently satisfy strong convexity. Instead, motivated by classical low-rank modeling approaches, we replace the non-convex rank constraint with a convex surrogate based on projection residual:
>     $$R(z) = \left\| (I - P_{U(c)})z \right\|^2,$$
>     which serves as a differentiable approximation of the low-rank prior. We treat strong convexity as an analytical assumption to ensure stable analysis of the reverse diffusion guidance, following RPCA (NeurIPS, 2019) and LRR (TPAMI).
>
> * **On Inter-class Subspace Orthogonality in Eq. (3):** Similarly, inter-class near-orthogonality is not a strict property of real data but an idealized separability objective inspired by OLE (CVPR, 2018). We adopt it as an optimization target rather than a hard constraint, encouraging lower intra-class variance and larger inter-class margins, thus improving discriminability under noise.
>
> ---
>
> [q4] — [Conflict between Semantic Prior and Visual Distribution]:
> To mitigate this, we propose a dynamic joint decision mechanism that ensures multi-modal consistency through the following two levels:
>
> * **Prototype-level Adaptive Fusion:** The refined prototype $p_{ref}$ is formulated as a *convex combination* of the initial geometric center $p_{\text{init}}$ and the semantic center $C_s$:
>     $$p_{ref} = \frac{\beta N}{\beta N + \gamma} p_{\text{init}} + \frac{\gamma}{\beta N + \gamma} C_s$$
>     where the weights are adaptively balanced by the number of support samples $N$ and hyperparameters $\beta, \gamma$. This ensures that the global prototype is governed by the source with higher relative confidence.
>
> * **Sample-level Joint Constraint:** For individual samples, we compute a joint weight $w_i$ by fusing geometric and semantic scores, balanced by a dynamic factor $\alpha$. Crucially, our **noise detection logic** follows a "dual-violation" principle: a sample $x_i$ is flagged as noise **only if** it contradicts both geometric and semantic priors simultaneously.
>
> In essence, semantic information is used as a regularizing auxiliary signal rather than a sole determinant. When the semantic prior is noisy, the geometric consistency constraint serves as a safeguard, mitigating its influence and preserving robust decision boundaries.

---

> > ### Author Rebuttal · Reviewer_q55V · 2026-04-02
> >
> > Thank you for the detailed rebuttal and the additional experimental results. I appreciate the clarifications, which resolve a large part of my concerns and help better explain several design choices in the paper. However, I still feel that some issues remain only partially addressed, particularly the more fundamental questions regarding the behavior of the proposed low-rank guidance in very high-noise settings and the practical validity of the theoretical assumptions. As a result, although I found the rebuttal helpful, it does not substantially change my overall assessment, and I will keep my original score.

---

> > > ### Author Response · Authors · 2026-04-02
> > >
> > > We thank the reviewer for the valuable comments and for the recognition of our method. Regarding the concern about the effectiveness of low-rank guidance under extreme noise conditions, we conduct additional analysis under a high-noise setting (sym_swap = 0.6). Overall, the results show that the low-rank mechanism remains the core component for recovering class-wise dominant structures. Even under high noise, it effectively compresses intra-class representations, enhances spectral concentration, and reduces reconstruction residuals, leading to more stable discriminative structures (see Table 1).
> > >
> > > Specifically, the effective rank decreases from 4.377 to 3.650, indicating stronger low-dimensional constraints on class-wise structure; the Top-3 energy increases from 0.886 to 0.929, suggesting more concentrated dominant features; and the residual ratio decreases from 0.114 to 0.071, indicating significantly reduced reconstruction error. These structural improvements ultimately lead to a significant improvement in classification performance (accuracy increases from 65.20 to 75.40).
> > >
> > > We are happy to address any further questions if needed.
> > >
> > > ---
> > >
> > >  (1) Low-rank Structure Recovery (noise = 0.6)
> > >
> > > | Method | Eff. Rank ↓ | Top-3 Energy ↑ | Residual ↓ | Accuracy ↑ |
> > > |---|---:|---:|---:|---:|
> > > | Diffusion w/o Low-Rank | 4.377 | 0.886 | 0.114 | 65.20 |
> > > | Diffusion w/ Low-Rank | 3.650 | 0.929 | 0.071 | 75.40 |

---

### Official Review · Reviewer_LGJ2 · 2026-03-17

**Soundness:** 3
**Presentation:** 3
**Significance:** 3
**Originality:** 3
**Overall Recommendation:** 4
**Confidence:** 3

**Summary:**

The paper introduces CRDProto, a noise-robust few-shot learning framework that assumes class features lie in a low-rank subspace and leverages rank deviations to identify noisy samples. It proposes a differentiable low-rank module (SVE) along with a rank-guided diffusion process to reconstruct corrupted features, followed by semantic-weighted prototype estimation. Experiments are conducted on MiniImageNet and TieredImageNet under few-shot learning settings.

**Compliance With Llm Reviewing Policy:**

Affirmed.

**Final Justification:**

My concerns are addressed, and I am increasing my scores to weak accept (4).

**Key Questions For Authors:**

Please refer to the weaknesses for the rebuttal.

**Limitations:**

I could not find the limitation section in the draft.

**Strengths And Weaknesses:**

Strengths:
- Novel conceptual design of a low-rank geometric view of noise in few-shot learning.
- The paper is generally well written and easy to follow.
- Comparison with several existing baselines

Weaknesses:
- The authors assume that within-class features are approximately rank-1; however, this assumption appears unrealistic for complex visual data and is not sufficiently validated. In practice, images within the same class can exhibit significant variability in factors such as pose, background, viewpoint, and scene context, which can introduce substantial intra-class diversity and lead to higher intrinsic dimensionality rather than a near rank-1 structure.
- Experiments are conducted on MiniImageNet and TieredImageNet, which may not fully capture the complexity of real-world scenarios. These benchmarks largely consist of images with a single, well-centered object and relatively clean backgrounds. However, real-world data often contains multiple objects, cluttered scenes, occlusions, and diverse contextual variations, making the setting significantly more challenging and potentially limiting the generalization of the proposed method.
- Noise is artificially injected, based on which results may not generalize to real-world noisy datasets.

---

> ### Author Rebuttal · Authors · 2026-03-31
>
> [q1,q2] — [Rank-1 Assumption on Complex Data]:
>
> We appreciate the reviewer’s insights. Our low-rank/rank-1 objective is a principled inductive bias for robust refinement, rather than a claim that raw visual data naturally satisfies this property. We utilize it as an active optimization target to align intra-class features toward shared semantic axes. By extracting stable commonalities through SVE and rank-guided diffusion, we progressively suppress irrelevant disturbances—such as background noise, viewpoint variations, and occlusions.
>
> Consequently, large initial intra-class variance does not contradict our hypothesis; the key is not the absolute magnitude of variance, but whether these variations can be concentrated into dominant semantic directions. Our results substantiate that low-rank guidance significantly compacts intra-class distributions and reduces distances, effectively enhancing consistency and filtering out non-semantic interference.
>
> **Low-rank dynamics summary (Meta-Dataset, noise = 0.4)**
>
> | Variant | Eff. Rank ↓ | Top-3 Energy Gain ↑ | Spectral Entropy ↓ | Residual Drop ↑ | Final Acc. ↑ |
> |----------|-------------|----------------------|---------------------|------------------|--------------|
> | Initial (No Refinement) | 4.38 | 0.00% | 1.48 | 0.00 | 52.47 |
> | Diffusion w/o Low-Rank | 3.92 | +4.12% | 1.35 | 0.04 | 54.40 |
> | Diffusion w/ Low-Rank | 2.85 | +18.53% | 0.92 | 0.18 | 62.93 |
> | **Full Method (SVE + Low-Rank)** | **2.14** | **+31.27%** | **0.65** | **0.26** | **74.12** |
>
> ---
>
> [q3] — [Benchmark Complexity & Real-World Generalization]:
>
> To ensure fair comparison, we strictly follow the protocols of recent SOTA methods. We adopt MiniImageNet and TieredImageNet as primary benchmarks, as these representative works predominantly utilize the 5-way 5-shot setting on them. Additionally, per the reviewer’s suggestion, we include Meta-Dataset results to further validate robustness under complex distributions.
>
> **Performance comparison under different noise levels across various backbones**
>
> | Method | Backbone | 0% Noise | 20% Noise | 40% Noise |
> |--------|----------|----------|-----------|-----------|
> | TSA (2022) | URL | 75.5 | 71.4 | 64.4 |
> | eTT (2022) | DINO | 80.9 | 76.7 | 67.5 |
> | AM3 (2019) | ResNet-12 | 74.1 | 68.4 | 62.5 |
> | FGFL (2023) | ResNet-12 | 80.1 | 74.8 | 69.9 |
> | MetaDiff (2024) | ResNet-12 | 74.5 | 69.2 | 63.5 |
> | SemFew-Trans (2024) | SwinT | 81.9 | 77.2 | 71.2 |
> | FedFSL-CFRD (2025) | ResNet-12 | 65.0 | 60.6 | 56.8 |
> | ECER-FSL (2025) | ResNet-12 | 84.4 | 78.1 | 73.3 |
> | ECER-FSL (2025) | Visformer-T | 85.2 | 79.8 | 74.3 |
> | LDC (2025) | CLIP | 85.2 | 79.2 | 74.2 |
> | DETA++ (2025) | SwinT | 86.5 | 80.9 | 77.3 |
> | ours | Swin-T | 91.9  | 89.9 | 76.9  |
> | ours | ViT-B/32 (CLIP) | 93.1  | 92.1 | 77.9 |
> | ours | ViT-S (DINO) | 90.0  | 88.0 | 74.2  |
> | ours | ResNet-50 | 89.1 | 88.1  | 74.1  |
>
> ---
>
> [q4] — [Synthetic vs. Real-world Noise Generalization]:
>
> To ensure fair comparison, we follow standard SOTA protocols on widely used benchmarks. Beyond conventional few-shot datasets, we conduct additional experiments on CIFAR-100N, Animal-10N, and Food-101N to evaluate robustness against real-world human and web noise. The results confirm our method’s stability, demonstrating that the proposed correction mechanism generalizes effectively from synthetic noise to realistic data corruption in practical applications.
>
> **Performance on real-world noisy datasets**
>
> | Dataset | #Classes | Accuracy (%) |
> |---------|----------|--------------|
> | CIFAR-100N | 100 | 58.7 ± 0.6 |
> | Animal-10N | 10 | 82.4 ± 0.4 |
> | Food-101N | 101 | 71.3 ± 0.5 |
>
> ---
>
> [q5] — [Missing Limitation Section]:
>
> We added a Limitation section addressing semantic manifold collapse under extreme noise ratios—a fundamental bottleneck in Noisy Few-Shot Learning (NFSL), where outliers become mathematically indistinguishable from true distributions. Reconstructing these collapsed manifolds via generative priors or self-supervised filtering remains a vital future direction. We hope our work provides a robust baseline for such unconstrained open-world scenarios.

---

> > ### Author Rebuttal · Reviewer_LGJ2 · 2026-04-03
> >
> > Thank you for your response. My concerns are addressed, and I am increasing my scores. Please add these additional details in the final version.

---

> > > ### Author Response · Authors · 2026-04-03
> > >
> > > Thank you very much for your positive feedback and recognition of our work. We will incorporate the suggested discussions and additional details into the final version to further improve the completeness and clarity of the manuscript.

---

### Decision · Program_Chairs · 2026-04-30

**Decision:**

Accept (regular)

**Comment:**

This paper proposes a differentiable low-rank module to detect and resolve noisy support examples for more robust few-shot learning. The paper is well-written and addresses a challenging problem within few-shot learning. Reviewers found the proposed approach to be novel and offer theoretical interpretability. Experiments compare against a large suite of baselines and were further strengthened during the rebuttal with additional comparisons on the Meta-Dataset benchmark. For these reasons, I recommend acceptance.